# Enhanced production of mesencephalic dopaminergic neurons from lineage-restricted human undifferentiated stem cells

Muyesier Maimaitili[1,2,12], Muwan Chen [1,2,12], Fabia Febbraro[2,3], Ekin Ucuncu[1,2], Rachel Kelly [1,2], Jonathan Christos Niclis[4], Josefine Rågård Christiansen [4], Noëmie Mermet-Joret[1,5,6], Dragos Niculescu[1,2,6], Johanne Lauritsen[1,2], Angelo Iannielli[7,8], Ida H. Klæstrup[1,2], Uffe Birk Jensen [2,3], Per Qvist [2,9,10,11], Sadegh Nabavi [1,5,6], Vania Broccoli [7,8], Anders Nykjær [1,2,6], Marina Romero-Ramos [1,2] & Mark Denham [1,2] ✉

Current differentiation protocols for generating mesencephalic dopaminergic (mesDA) neurons from human pluripotent stem cells result in grafts containing only a small proportion of mesDA neurons when transplanted in vivo. In this study, we develop lineage-restricted undifferentiated stem cells (LR-USCs) from pluripotent stem cells, which enhances their potential for differentiating into caudal midbrain floor plate progenitors and mesDA neurons. Using a ventral midbrain protocol, 69% of LR-USCs become bona fide caudal midbrain floor plate progenitors, compared to only 25% of human embryonic stem cells (hESCs). Importantly, LR-USCs generate significantly more mesDA neurons under midbrain and hindbrain conditions in vitro and in vivo. We demonstrate that midbrain-patterned LR-USC progenitors transplanted into 6-hydroxydopamine-lesioned rats restore function in a clinically relevant non-pharmacological behavioral test, whereas midbrain-patterned hESC-derived progenitors do not. This strategy demonstrates how lineage restriction can prevent the development of undesirable lineages and enhance the conditions necessary for mesDA neuron generation.

Mesencephalic dopaminergic (mesDA) neurons develop from the ventral midbrain of the neural tube. The morphogen sonic hedgehog (SHH) and members of the WNT family are instrumental in their specification due to their essential role in establishing the dorsal-ventral and anterior-posterior (A-P) axes of the embryo, respectively[1,2]. For the production of midbrain dopaminergic (DA) neurons, high SHH signaling from the notochord is required to specify ventral neural epithelial cells in the neural plate to a floor plate identity[3], and graded WNT signaling emanating from the posterior regions of the embryo is required to specify anterior neuroectoderm to a midbrain identity[4]. At

[1]Danish Research Institute of Translational Neuroscience (DANDRITE), Nordic EMBL Partnership for Molecular Medicine, Aarhus University, 8000C Aarhus, Denmark. [2]Department of Biomedicine, Aarhus University, 8000C Aarhus, Denmark. [3]Department of Clinical Genetics, Aarhus University Hospital, 8200 Aarhus, Denmark. [4]Cell Therapy Discovery, Cell Therapy R&D, Novo Nordisk, Måløv, Denmark. [5]Department of Molecular Biology and Genetics, Aarhus University, 8000C Aarhus, Denmark. [6]Center of Excellence PROMEMO, Aarhus University, Aarhus, Denmark. [7]Stem Cell and Neurogenesis Unit, Division of Neuroscience, IRCCS San Raffaele Scientific Institute, 20132 Milan, Italy. [8]CNR Institute of Neuroscience, 20129 Milan, Italy. [9]Lundbeck Foundation Initiative for Integrative Psychiatric Research, iPSYCH, 8000C Aarhus, Denmark. [10]Centre for Integrative Sequencing, iSEQ, Aarhus University, 8000C Aarhus, Denmark. [11]Centre for Genomics and Personalized Medicine, CGPM, Aarhus University, 8000C Aarhus, Denmark. [12]These authors contributed equally: Muyesier Maimaitili, Muwan Chen. ✉e-mail: mden@dandrite.au.dk

later stages in development, neural progenitors in the midbrain region receive WNT1 and FGF8 signals from the isthmic organizer, which refines the patterning of the cells to a caudal location within the mesencephalon[5,6]. Recapitulating these developmental steps in vitro with human pluripotent stem cells (hPSCs) is the goal of stem cell transplantation therapies for Parkinson's disease[7].

In stem cell differentiation protocols, early application of high concentrations of SHH is necessary to specify neural progenitor cells to a floor plate identity[8]. However, along the A-P axis (also referred to as the rostral-caudal axis), the process governing specification to a caudal midbrain identity is more complex. A titrated WNT concentration within a precise range can specify anterior neuroectoderm progenitors to a caudal midbrain identity[9–11]. High concentrations of WNT result in the production of hindbrain cell types, and lower concentrations result in an anterior midbrain or diencephalic identity. Timed delivery of FGF8 or sequential exposure to high levels of WNT has also been shown to enhance specification to a caudal midbrain identity[11–13]. Furthermore, dual canonical and non-canonical WNT activation can improve caudal midbrain patterning[14], and recently retinoic acid (RA) was used in place of WNT and FGF8 to caudalize neuroectoderm progenitors to a midbrain fate[15]. Among these protocols, long-term in vitro differentiation to DA neurons has been achieved, with a wide range of TH-positive percentages being reported[9,16]. In most studies, the overall yield of mesDA neurons following transplantation in vivo is low or highly variable, indicating heterogeneity within the progenitor population[12,13,16,17].

Assessing midbrain specification has commonly involved the detection of LMX1A/FOXA2 double-positive progenitors, which represent the ventral diencephalon and mesencephalon[18,19], and high percentages (>80%) of these cells have been reported by several laboratories[9,16]. However, to distinguish diencephalic cells from the mesencephalon additional markers such as EN1 are required. *En1* is expressed early in the midbrain and rhombomere (r)1[20], and at later stages, the anterior limit shifts posteriorly to the caudal midbrain[19]. Recently, high *EN1* expression in progenitor populations has been shown to correlate with graft function[12], and several laboratories have shown EN1 in hPSC-derived midbrain progenitors and its coexpression with LMX1A and OTX2 (a forebrain/midbrain marker)[12,13]. However, few laboratories have quantified hPSC-derived progenitor using combined markers such as OTX2 and EN1[11,21], which together precisely define the caudal midbrain (Fig. 1a). Importantly, at later stages in development EN1 can be used to distinguish mesDA populations from floor plate-derived neurons in anterior populations, some of which express TH, FOXA2, and NR4A2 but are not mesDA neurons[19,22]. Hence, quantifying the percentage of EN1/TH coexpressing neurons in vitro is required to accurately discern the presence of non-dopaminergic lineages[11].

During development, progenitor cells employ gene regulatory networks (GRNs) to interpret morphogen gradients and regulate cell fate[23]. Within the developing neural tube, *Otx2* is expressed in anterior regions, whereas *Gbx2* is expressed in posterior regions at early stages of development. The midbrain-hindbrain boundary (MHB) demarcates the *Otx2* and *Gbx2* boundary and the location of the isthmic organizer[6]. At this border, *Otx2* and *Gbx2* establish a separate network of transcription factors that maintain the position of the isthmic organizer and assist in patterning the surrounding region[5,24–26]. Alterations in the expression levels of transcription factors in networks can result in the expansion or loss of specific brain regions[27]. Transcription factor networks have been altered by forced expression of lineage-determining transcription factors, such as LMX1A, which accelerates the differentiation of mouse embryonic stem cells (mESCs) and human embryonic stem cells (hESCs) into DA cells[28,29], and forced expression of *GLI1* in hESC-derived neural progenitors can generate floor plate cells[30]. Conversely, in developing embryos the ablation of transcription factors can result in the loss of specific cell populations. Along the A-P axis, deletion of *Otx2* in embryos results in the loss of forebrain and

midbrain structures[31,32]. Null mutations in all three *Cdx* family members result in the loss of spinal cord cell types below the preoccipital level due to disruption of central and posterior *Hox* gene expression and prevention of neuromesodermal progenitor (NMP) formation[33–35]. Interestingly, null mutations in *Gbx2* result in the posterior shift of the MHB and the expansion of the midbrain at the expense of r1-3[36]. These studies demonstrate that ablation of transcription factors that control cell fate can lead to the activation of altered GRNs and the respecification or expansion of alternate populations. It is therefore possible that transcription factor determinants in GRNs that are involved in lineage choices can be disrupted to control cell fate and bias the differentiation of hPSCs toward a mesencephalic neuron identity.

In this study, we used a gene knockout approach to restrict cell fate and prevent the differentiation of non-DA cell lineages with the aim of enhancing differentiation to mesDA neurons. Specifically, we focused on the early developmental stages when major lineage choices are made and identified the transcription factor determinates that are critical for those lineages but not required for a mesDA fate. By inducing loss-of-function mutations in lineage determinant genes expressed in non-DA lineages, we generated stem cells that could be expanded in the undifferentiated pluripotent state and were restricted in their potential when differentiated. We named these lineage-restricted undifferentiated stem cells (LR-USCs). Importantly, we show that upon differentiation LR-USCs favored the differentiation down a mesDA lineage. To precisely identify the caudal midbrain floor plate, we developed a FACS panel to quantify cells quadruple positive for FOXA2/LMX1A/OTX2/EN1. Our results demonstrated that lineage restriction can be successfully used to control cell fate and enhance the generation of functional mesDA neurons, and the development of these cells is less reliant on specific concentrations of extrinsic factors.

## Results

### Midbrain cell types are preferentially generated from pluripotent stem cells containing biallelic null mutations in *GBX2*, *CDX1*, *CDX2*, and *CDX4*

To restrict the differentiation of hPSCs and guide them toward a mesDA neuron identity, we introduced null mutations in transcription factors that regulate cell fate along the A-P axis. First, we investigated whether a biallelic null mutation in *GBX2* in hESCs (H9 cells) results in an increase in the production of midbrain cell types when the cells are differentiated under conditions known to produce hindbrain and spinal cord cells. We generated a *GBX2*[-/-] hESC line by introducing indels into the coding sequence (Supplementary Fig. 1). In the undifferentiated state, the *GBX2*[-/-] cells were morphologically indistinguishable from control hESCs and capable of differentiating into neural progenitors. Using our previously published protocol for generating caudal neural progenitors (CNPs)[37], we differentiated *GBX2*[-/-] cells into CNPs for four days in vitro (DIV) and compared these cells to H9 (control) CNPs (Fig. 1b). Indeed, when differentiated in the presence of a GSK3B inhibitor (GSK3i; CHIR99021) at a concentration known to give rise to hindbrain and spinal cord cells (3 μM), there was a small but significant increase in the expression of the forebrain/midbrain marker *OTX2* (LogFC to hESC = H9: 0.003; *GBX2*[-/-]: 0.083; *P* = 0.0005; Fig. 1c, d) and a significant reduction in the transcript level of *CDX2* in the *GBX2*[-/-] cells compared to H9 cells (LogFC to hESC = H9: 8871.79; *GBX2*[-/-]: 3160.57; *P* = 0.006; Fig. 1e). Despite the increase in *OTX2* expression, we observed that 0.41% of cells were OTX2-positive and that 94.73% of cells were CDX2-positive (Fig. 1c and Supplementary Fig. 1i, j).

Based on the results obtained using the *GBX2*[-/-] line, we next aimed to further restrict the potential of cells along the A-P axis by knocking out *CDX* family members. CDX2 is an upstream regulator of central and posterior *Hox* genes and acts as a key determinant of spinal cord fate through its regulation of axial elongation[33,38]. Triple biallelic null mutations in all three *Cdx* genes (*Cdx1*, *Cdx2*, and *Cdx4*) result in severe

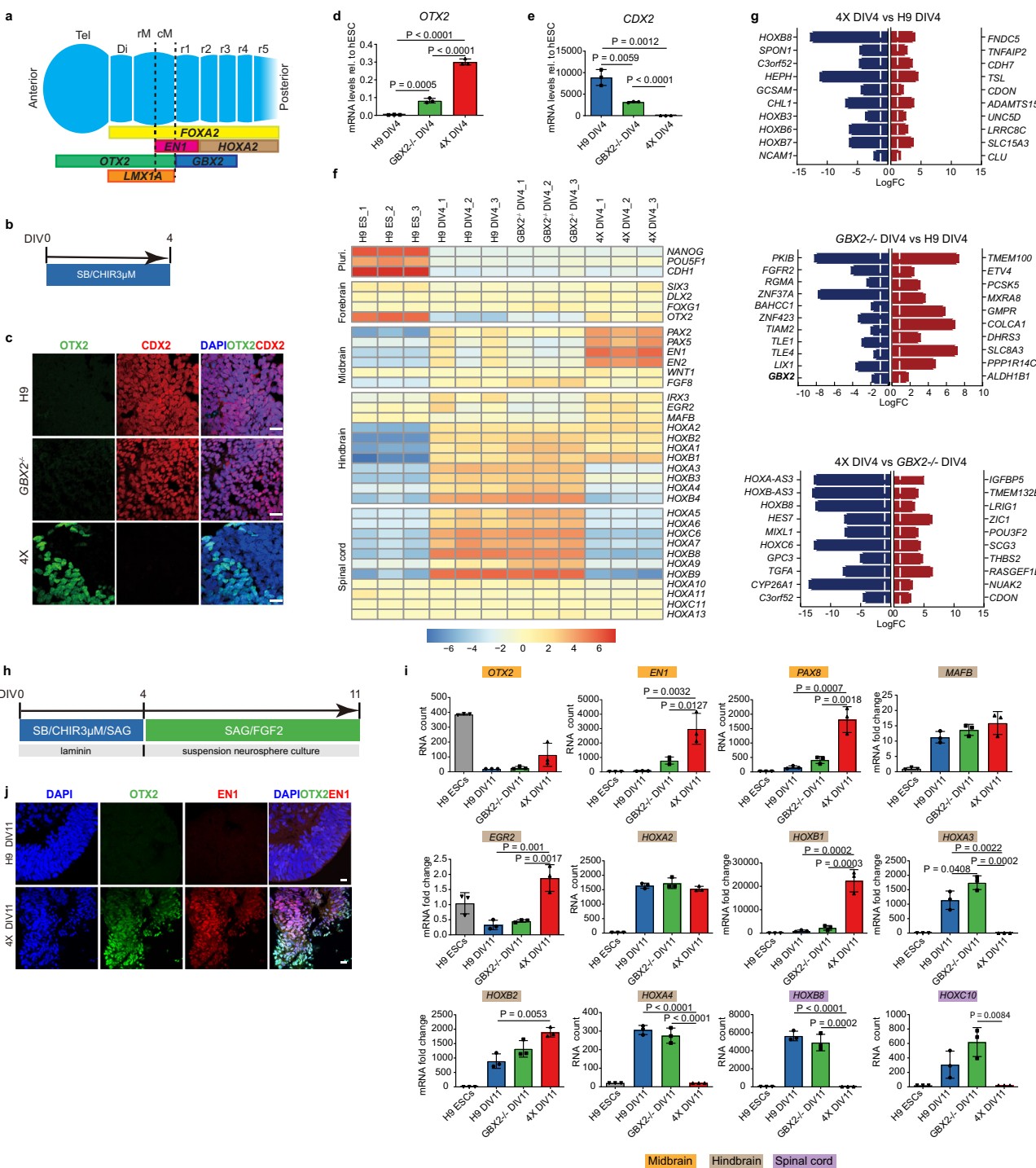

**Fig. 1 | Differentiation of GBX2-/- and 4X cells using a caudal neural progenitor protocol. a** Schematic gene expression profile of genes along the A-P axis that define regions from the telencephalon (Tel) to rhombomere (r)5 during embryonic development. **b** Schematic diagram of the DIV4 CNP differentiation protocol. **c** Representative immunofluorescence images of H9 and GBX2[-/-] cells at DIV4 showing OTX2-/CDX2+ cells. A few 4X cells were positive for OTX2, but no cells were positive for CDX2. Scale bars, 20 μm. QPCR analysis of *OTX2* (**d**) and *CDX2* (**e**) expression in H9, *GBX2[-/-]* and 4X cells at DIV4. The data are presented as the mean ± SD; *n* = 3 biological replicates. One-way ANOVA showed statistical significance, and then an unpaired *t* test comparing two groups was performed. **f** Heatmap of the expression of pluripotent and neural genes representing the forebrain, midbrain, hindbrain and spinal cord regions in H9, *GBX2[-/-]* and 4X cells at DIV4. **g** The top 10 downregulated (blue) and upregulated (red) genes (and

additional selected gene in bold) between 4X and H9 cells, *GBX2[-/-]* and H9 cells, and 4X and *GBX2[-/-]* cells at DIV4. The threshold bar (white line) indicates a fold change of ±2. **h** Schematic diagram of the DIV11 CNP differentiation protocol. **i** RNA expression analysis of the midbrain genes (orange) *OTX2, EN1* and *PAX8*; the hindbrain genes (gray) *MAFB, EGR2, HOXA2, HOXB1, HOXA3, HOXB2,* and *HOXA4*; and the spinal cord genes (purple) *HOXB8* and *HOXC10*. Nanostring data shown as RNA count and QPCR as fold change. The data are presented as the mean ± SD; *n* = 3 biological replicates. One-way ANOVA followed by Tukey's multiple comparisons test. **j** Representative immunofluorescence analysis of OTX2/EN1 double-positive cells among DIV11 4X cells. No OTX2/EN1 double-positive cells were detected among H9 cells. Scale bars, 10 μm. Diencephalon: Di, rostral midbrain: rM, caudal midbrain: cM, caudal neural progenitor: CNP. Source data are provided as a Source Data file.

truncation of the spinal cord below the postoccipital level and prevent the formation of NMPs[34,35]. Because loss of *Cdx1/2/4* in mice causes posterior truncation, we chose to disrupt the *CDX* family of genes in addition to knocking out *GBX2*. We generated biallelic null mutations in all three *CDX* family members, *CDX1/2/4*, by targeting their DNA binding domains (Supplementary Fig. 2). In the resulting knockout cell line, we analyzed the exons for off-target activity and detected no changes; furthermore, we observed no aneuploidy (Supplementary Table 1 and Supplementary Fig. 2). The resulting *GBX2^/-^CDX1/2/4^/-^* cells (hereafter referred to as 4X cells) were differentiated for four DIV using the CNP protocol (Fig. 1b). As expected, we did not detect *CDX2* transcripts or CDX2 expression in 4X cell-derived CNPs (Fig. 1c, e). Strikingly, at four DIV, the 4X neural progenitors showed a significant increase in *OTX2* transcript levels compared to H9 and *GBX2^/-^* derived CNPs (LogFC to hESC = H9: 0.0032; *GBX2^/-^*: 0.083; 4X: 0.301; $P < 0.0001$ for both 4X vs. H9 or *GBX2^/-^*), and we could readily identify OTX2-positive cells (Fig. 1c, d).

To elucidate the effects on gene expression in more detail, we performed RNA sequencing of CNPs derived from all three cell lines (H9, *GBX2^/-^* and 4X cells). We assessed *HOX* gene expression profiles and found that the 4X CNPs showed a significant reduction in the expression of posterior *HOX* genes, compared to H9 CNPs, beginning with *HOXA3* and moving caudally (*HOXA3*, $P = 1.7 \times 10^{-9}$; *HOXA5*, $P = 1.39 \times 10^{-7}$; HOXA7, $P = 1.85 \times 10^{-5}$; *HOXA9*, $P = 1.2 \times 10^{-5}$; *HOXA10*, $P = 0.0001$; Fig. 1f). These results indicate that 4X cells were unable to generate progenitor cell types caudal to r4[39]. We next questioned whether there were changes in the expression of anterior genes. First, we examined the expression of forebrain genes in 4X CNPs and observed no change in the expression of *SIX3*, *DLX2* and *FOXG1* (Fig. 1f). However, the transcript levels of the forebrain/midbrain gene *OTX2* were significantly increased in 4X CNPs compared to H9 and *GBX2^/-^* CNPs (H9, LogFC = 6.83, $P = 0.001$; *GBX2^/-^*, LogFC = 3.92, $P = 0.003$, respectively; Fig. 1f). The expression of the midbrain genes *PAX2* and *EN1* was also significantly increased in 4X CNPs compared to H9 CNPs (*PAX2*, LogFC = 2.63, $P = 0.005$; *EN1*, LogFC = 4.76, $P = 0.007$) and *GBX2^/-^* CNPs (*PAX2*, LogFC = 3.49, $P = 6.46 \times 10^{-6}$; *EN1*, LogFC = 7.08, $P = 5.1 \times 10^{-5}$; Fig. 1f). Interestingly, *GBX2^/-^* cells showed a reduction in the expression of anterior hindbrain genes, such as *EGR2* (LogFC = −3.86; also known as *KROX20)* and *MAFB* (LogFC = −0.67), which was in line with reports showing that disruption of *Gbx2* in mice causes loss of r1-3[36]. In contrast, the expression of *MAFB* significantly increased in 4X cells (*MAFB*, LogFC = 2.49, $P = 3.93 \times 10^{-6}$), which is in accordance with the loss of *CDX1* and posterior *HOX* expression[40].

Analysis of differentially expressed genes among the three groups showed that the top significantly downregulated genes in 4X cells compared to H9 and *GBX2^/-^* cells included posterior *HOX* genes (Fig. 1g). Furthermore, the expression of *CYP26A1*, which is involved in RA metabolism and is induced by CDX2, was significantly downregulated in 4X cells compared to *GBX2^/-^* cells (LogFC = −13.43, $P = 7.6 \times 10^{-16}$). A comparison of *GBX2^/-^* cells and H9 cells showed that knockout of *GBX2* alone resulted in a significant decrease in the transcription levels of the Groucho corepressor proteins TLE1 (LogFC = −2.86, $P = 9.96 \times 10^{-12}$) and TLE4 (LogFC = −1.54, $P = 1.09 \times 10^{-11}$), which interact with GBX2 to repress OTX2[41]. Overall, knockout of *GBX2* resulted in disruption of anterior hindbrain patterning, and 4X cells showed that further loss of CDX family members caused a posterior limitation of the CNS equivalent of r4. Consequently, when 4X is differentiated using a protocol that produces posterior hindbrain and spinal cord cells, it cannot generate these cell types. Instead, 4X cells adopt an alternate fate of a more anterior identity, specifically midbrain cells or the remaining region of the hindbrain not regulated by the four genes. This is supported by our results showing significantly higher expression of midbrain and anterior hindbrain genes (Fig. 1f, g).

To further explore the differentiation potential of the 4X cells, we extended the duration of differentiation to 11 and 32 DIV and added smoothened agonist (SAG) to ventralize the cells (Fig. 1h and Supplementary Fig. 3). At 11 DIV, we found that the expression of the midbrain gene *PAX8* was significantly upregulated in 4X cells compared to H9 and *GBX2^/-^* cells (RNA count, H9: 158.6; *GBX2^/-^*: 414.4; 4X: 1813.0; $P = 0.0007$, $P = 0.002$, respectively; Fig. 1i). The expression of *EN1*, which spans the caudal midbrain and r1 during development[42], was also significantly upregulated in 4X cells compared to H9 and *GBX2^/-^* cells (RNA count = H9: 66.6; *GBX2^/-^*: 784.8; 4X: 2985.4; $P = 0.003$ and $P = 0.01$, respectively; Fig. 1i). The expression of the hindbrain gene *EGR2*, which is expressed in r3 and r5[43], was significantly upregulated (LogFC = H9: 0.33; *GBX2^/-^*: 0.46; 4X: 1.88; $P = 0.001$ and $P = 0.002$, respectively; Fig. 1i), and the expression of *MAFB*, a marker of r5 and r6[43], was not significantly altered. Similar changes were also found at 32 DIV. Additionally, we found that at 32 DIV the transcript levels of *OTX2* were significantly higher in the 4X cells than in the H9 cells (LogFC = 6.8, $P = 7.82 \times 10^{-10}$); this change in *OTX2* at 32 DIV was similar to what was observed after differentiation for four DIV (Supplementary Fig. 3).

Upon examination of *HOX* expression profiles, we found that central and posterior *HOX* genes beginning with *HOXA3* and moving posteriorly were absent in 4X cells (Fig. 1i). The expression of *HOXA2*, which is expressed throughout the hindbrain (except for r1), was maintained in the 11 DIV 4X cell-derived CNPs; however, the expression of the anterior *HOX* genes *HOXB2* and *HOXB1* was significantly upregulated in 4X cells compared to H9 cells (LogFC = H9: 893.4; 4X: 1893.3; $P = 0.005$ for *HOXB2*, and H9: 965.0; 4X: 22377.6; $P = 0.0002$ for *HOXB1*; Fig. 1i), suggesting a compensatory shift in the population to a more anterior identity. Immunocytochemical analysis confirmed the change that we observed at the transcript level. We identified caudal midbrain progenitors, i.e., OTX2/EN1 double-positive cells, among 4X cells at 11 DIV, but not among H9 cells at 11 DIV (Fig. 1j). These results indicate that under caudalizing conditions, 4X cells did not produce spinal cord progenitors and showed a restricted *HOX* expression profile up to r4, which was also confirmed when the differentiation was extended to 32 DIV (Supplementary Fig. 3). These results support the notion that 4X cells are lineage restricted and preferentially adopt a midbrain or anterior hindbrain identity under conditions that usually give rise to caudal hindbrain and spinal cord cell types.

## LR-USCs efficiently generate caudal midbrain floor plate progenitors and mesDA neurons

Our main objective was to determine whether LR-USCs can more efficiently generate mesDA neurons than hPSCs. Thus, we performed a benchmarking study and compared H9 and 4X cells using a mesDA differentiation protocol that is in use in a clinical trial (Fig. 2a)[16]. This protocol is known to require adjustments to the concentration of GSK3i between cell lines; therefore, we used GSK3i at a concentration of 0.6 μM to 0.8 μM. Previous reports have shown that following this protocol, over 80% of cells express LMX1A/FOXA2[16]. However, LMX1A/FOXA2 double-positive progenitor cells include progenitors for both the mesDA neurons and subthalamic nucleus[18,19]. To accurately assess the percentage of caudal midbrain floor plate progenitors, we included OTX2 and EN1 in our quantitative expression analysis. EN1 is specific to the caudal midbrain and r1[19], and OTX2 is expressed in the forebrain and midbrain and absent from the hindbrain. We developed a flow cytometry panel to simultaneously examine the protein expression of the four transcription factors FOXA2, OTX2, LMX1A, and EN1 (Fig. 2b; Supplementary Fig. 4). We first assessed the percentage of LMX1A/FOXA2 double-positive cells and found that, as in other studies, our control H9 cells and lineage-restricted 4X cells both generated a high percentage of LMX1A/FOXA2 positive cells (0.6 μM H9: 63.4% ± 13.5, 4X: 61.0% ± 8.2; 0.7 μM H9: 75.4% ± 3.0, 4X: 74.2% ± 2.8; 0.8 μM H9: 69.6% ± 9.9, 4X: 75.3% ± 3.0; Fig. 2c, d). Second, we examined the percentage of OTX2/EN1 double-positive cells and found that 4X cells produced a significantly higher percentage of these cells (0.6 μM H9: 21.7% ± 9.2, 4X: 40.3% ± 13.9, $P = 0.0553$; 0.7 μM H9: 25.7% ± 7.2, 4X:

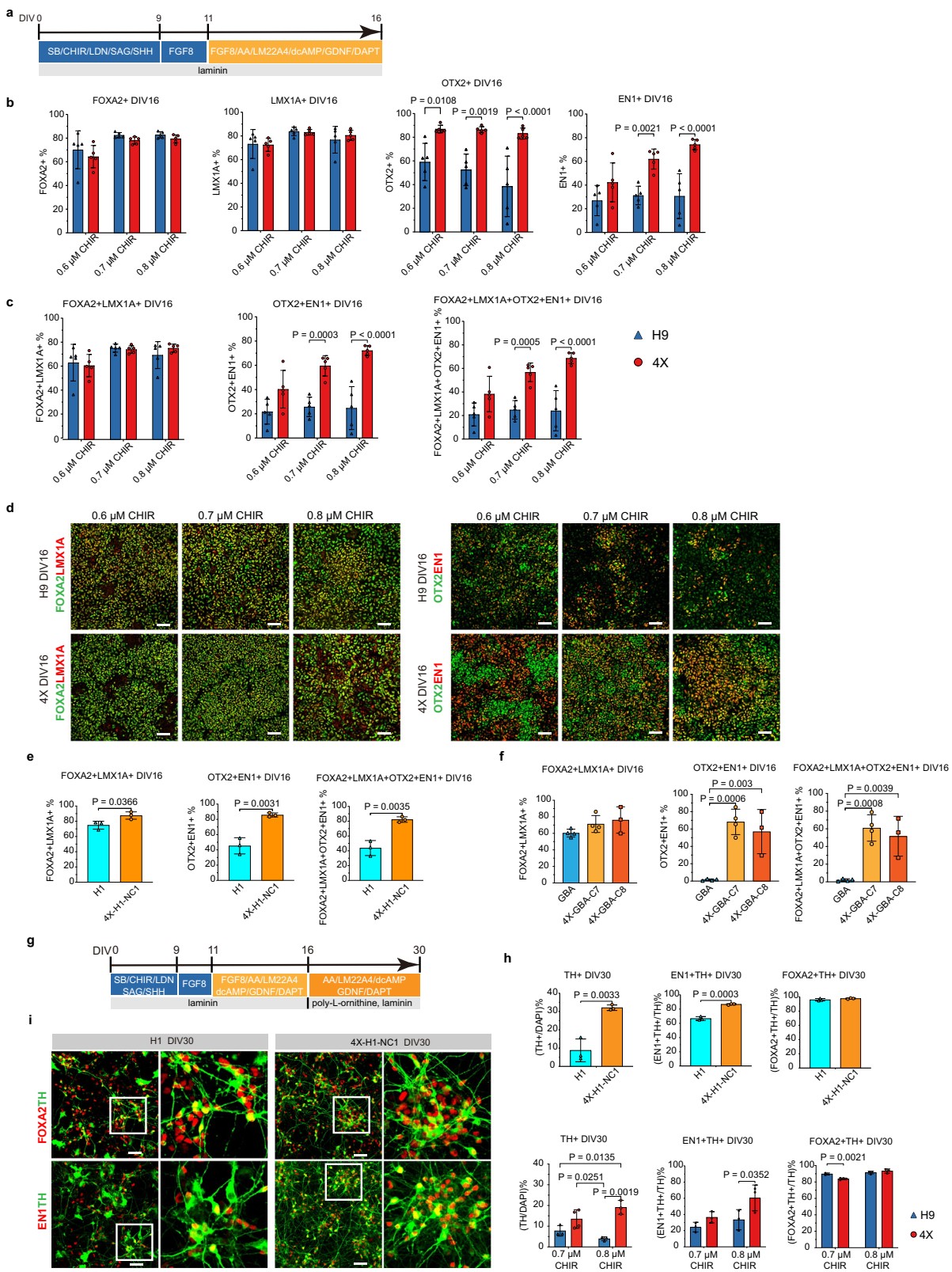

59.6% ± 7.5, *P* = 0.0003; 0.8 μM H9: 24.7% ± 15.8, 4X: 72.2% ± 3.5, *P* < 0.0001; Fig. 2c, d). We next quantified the percentage of cells expressing all four transcription factors and found that 4X cells maintained a higher percentage of FOXA2/LMX1A/OTX2/EN1 quadruple-positive cells than H9 cells across all GSK3i concentrations (0.6 μM H9: 20.9% ± 8.7, 4X: 38.2% ± 13.5, *P* = 0.0661; 0.7 μM H9: 24.8% ± 7.0, 4X: 56.7% ± 7.1, *P* = 0.0005; 0.8 μM H9: 24.0% ± 15.3, 4X:

68.7% ± 4.1, *P* < 0.0001; Fig. 2c). These results illustrate that a complex set of markers is required to define the caudal midbrain floor plate with precision, and that lineage-restricted 4X cells are significantly more efficient at producing both OTX2/EN1 double-positive and FOXA2/ LMX1A/OTX2/EN1 quadruple-positive cells than controls and robustly maintain a higher efficiency across a broader range of GSK3i concentrations.

**Fig. 2 | Differentiation into ventral midbrain progenitors and mesDA neurons.**
**a** Schematic diagram of the DIV16 midbrain differentiation protocol using different concentrations of GSK3i ranging from 0.6 μM to 0.8 μM. Flow cytometry analysis of the percentage of (**b**) FOXA2 + , LMX1A + , OTX2+ and EN1+ cells, and (**c**) FOXA2 + LMX1A + , OTX2 + EN1 + , FOXA2 + LMX1A + OTX2 + EN1+ cells among H9 and 4X at DIV16 after administration of GSK3i at concentrations ranging from 0.6 μM to 0.8 μM. The data are presented as the mean ± SD; n = 5, from independent experiments. Two-way ANOVA followed by Sidak's multiple comparisons test.
**d** Representative immunofluorescence analysis of FOXA2, LMX1A, OTX2, and EN1 expression in H9 and 4X cells treated with GSK3i at concentrations of 0.6 μM, 0.7 μM and 0.8 μM. Scale bars, 50 μm. **e, f** Flow cytometry analysis of FOXA2 + LMX1A + , OTX2 + EN1 + , FOXA2 + LMX1A + OTX2 + EN1+ cells among H1, 4X-H1-NC1, GBA, 4X-GBA-C7, and 4X-GBA-C8 cells at DIV16 when GSK3i was 0.8 μM. The

data are presented as the mean ± SD; (H1, 4X-H1-NC1, 4X-GBA-C8: n = 3; GBA, 4X-GBA-C7: n = 4) from independent experiments. An unpaired two-tailed t test was used to compare groups in (**e**) and one way ANOVA with Dunnett's test for (**f**).
**g** Schematic diagram of the DIV30 midbrain differentiation protocol.
**h** Quantification of TH/DAPI, (FOXA2 + TH + )/TH, (EN1 + TH + )/TH-percentage cells among H9, 4X, at DIV30 when GSK3i were 0.7 μM and 0.8 μM, and H1 and 4X-H1-NC1 at DIV30 when GSK3i was 0.8 μM. The data are presented as the mean ± SD; (n = 3 all conditions excect n = 4 for 4X when GSK3i was 0.8 μM), from independent experiments. Two-way ANOVA followed by Sidak's multiple comparisons test was used for H9 and 4X. An unpaired two-tailed t test was used to compare H1 and 4X-H1-NC1 groups. **i** Representative images for FOXA2, TH, EN1 staining at DIV30. Scale bars: 50 μm. Source data are provided as a Source Data file.

To investigate how lineage restriction affects other cell lines, we used H1 hESCs[44] and DANi002C iPSCs (heterozygous for the GBA L444P variant, hereafter referred to as GBA cells)[45]. We generated one H1 clonal cell line with knockout of *GBX2* and *CDX1/2/4* (called 4X-H1-NC1 cells) and two clones from DANi002C in which the same genes were knocked out (called 4X-GBA-C7 and 4X-GBA-C8 cells; Supplementary Fig. 5). The three cell lines and their PSC control lines were differentiated for 16 DIV using 0.8 μM of GSK3i. All lines generated a high percentage of FOXA2/LMXA1 double-positive cells (H1: 74.9% ± 5.1; 4X-H1-NC1: 87.5% ± 4.8; GBA: 60.4% ± 4.7; 4X-GBA-C7: 71.3% ± 10.2; 4X-GBA-C8: 76.1% ± 15.9; Fig. 2e, f; Supplementary Fig. 6a), and only 4X-H1-NC1 was significantly higher than its control (P < 0.05; Fig. 2e). However, all 4X lines produced a significantly higher percentage of OTX2/EN1 double-positive cells then their PSC control (P < 0.01 for 4X-H1-NC1, P < 0.001 for 4X-GBA-C7, and P < 0.01 for 4X-GBA-C8; Fig. 2e, f). Importantly, we examined the percentage of FOXA2/LMX1A/OTX2/EN1 quadruple-positive cells and again found that all 4X lines generated significantly higher percentages then their PSC control (P < 0.01 for 4X-H1-NC1, P < 0.001 for 4X-GBA-C7, and P < 0.01 for 4X-GBA-C8; Fig. 2e, f). Altogether, these results show that LR-USCs produced by knockout of *GBX2* and *CDX1/2/4* significantly enhances caudal midbrain floor plate specification independent of the genetic background.

We next extended the differentiation of H9, H1, 4X, and 4X-H1-NC1 cells to 30 DIV (Fig. 2g) and quantified the percentage of TH-positive and FOXA2/TH double-positive cells in the cultures (Fig. 2h; Supplementary Fig. 6b). Given the importance of EN1 for the survival of DA neurons, and recent reports showing the correlation of *EN1* expression with the proper identity of DA neurons and graft function[12], we also assessed the percentage of EN1/TH double-positive cells. We found that 4X-H1-NC1 cells generated a significantly higher percentage of TH-positive cells than H1 cells (H1: 8.8% ± 6.2; 4X-H1-NC1: 32.1% ± 1.5; P = 0.0033; Fig. 2h, i). Within these TH-positive cells, 4X-H1-NC1 cells generated a significantly higher percentage of EN1/TH double-positive cells (H1: 66.7% ± 2.7; 4X-H1-NC1: 86.6% ± 0.9; P = 0.0003; Fig. 2h, i), and, as expected, there was no significant difference in the percentage of FOXA2/TH-positive cells produced by 4X-H1-NC1 cells and H1 cells (H1: 96.0% ± 1.7; 4X-H1-NC1: 97.7% ± 0.9; P = 0.1937; Fig. 2h). Similar results were observed for H9 and 4X cells at GSK3i concentrations of 0.7 μM and 0.8 μM. Higher numbers of TH-positive cells were observed in 4X cells compared to H9 cells (0.7 μM H9: 7.8% ± 2.6, 4X: 13.5% ± 4.5, P = 0.0962; 0.8 μM H9: 3.9% ± 0.9, 4X: 19.1% ± 3.3; P = 0.0019; Fig. 2h and Supplementary Fig. 6b). Within these TH cells, a significantly higher percentage of EN1/TH double-positive cells were identified in the 4X cultures compared to H9 when the GSK3i concentration was 0.8 μM (0.7 μM H9: 24.4% ± 5.9, 4X: 36.5% ± 6.9, P = 0.3913; 0.8 μM H9: 33.5% ± 12.5, 4X: 60.4% ± 15.9; P = 0.0352; Fig. 2h and Supplementary Fig. 6b).

## LR-USCs generate midbrain progenitors when differentiated under hindbrain conditions

To further test the lineage restriction of the 4X cells and their ability to generate DA neurons, we modified the mesDA neuron protocol to

generate anterior hindbrain cells by increasing the concentration of GSK3i. In the first experiments we used 3 μM which in hESCs primarily generate posterior hindbrain and spinal cord cells that are regulated by CDX genes (Fig. 1f). In this experiment we patterned the cells to an anterior hindbrain region which is controlled by GBX2. Thus, we used a GSK3i concentration of 1 μM (Fig. 3a). We reasoned that a hindbrain differentiation protocol should yield few midbrain cells from hESCs, and would be a decisive proof-of-concept strategy for assessing lineage restriction towards the mesencephalon fate. Using single-cell RNA-sequencing, we determined the cell types that were produced by H9 and 4X cells following 16 DIV. Dimension reduction was performed by uniform manifold approximation and projection (UMAP), and a significant separation between 4X and H9 cells was observed across clusters (chi-square, P < 0.0001); moreover, there was a difference in the cell types produced by the two cell lines at 16 DIV (Fig. 3b–g and Supplementary Fig. 7). This separation coincided with a marked shift in the distribution along the A-P axis. Based on the expression of *OTX2* and *EN1*, we divided the A-P axis into four domains (rostral, *OTX2*-positive/*EN1*-negative; caudal midbrain, *OTX2*-positive/*EN1*-positive; r1, *OTX2*-negative/*EN1*-positive; and posterior, *OTX2*-negative/*EN1*-negative) (Fig. 3f). 4X cells produced all four populations, with the smallest being the most rostral population (Fig. 3g). The caudal midbrain is the region where DA neurons of the substantia nigra develop, and 35.2% of 4X cells could be assigned to this region. Whereas, only 0.6% of the cells produced by H9 cells expressed caudal midbrain markers, and 97.5% of the cells were of the posterior population (*OTX2*-negative/*EN1*-negative; Fig. 3g). Overall, 4X cells preferentially generated *EN1*-positive cells (54.3%) spanning the midbrain and hindbrain, whereas the majority (97.5%) of H9 cells were classified as hindbrain cell types. Analysis of the expression of the ventral patterning gene *FOXA2* showed that it was highly expressed in both H9 and 4X cells, indicating ventralization to a floor plate identity was unperturbed (Fig. 3f).

Upon further examination of caudal midbrain cells by graph-based clustering, we identified two clusters (clusters 2 and 5) enriched in caudal midbrain floor plate progenitors expressing *FOXA2*, *OTX2*, *LMX1A*, and *EN1* (Fig. 3b, d, e). Two additional midbrain floor plate clusters (clusters 4 and 6) were identified; these populations expressed *FOXA2*, *OTX2*, and *EN1* but lacked *LMX1A*, indicating that they were a lateral floor plate population. Clusters 4 and 5 were in a proliferative state, as revealed by the expression of *MKI67* and *TOP2A* and by analysis of cell-cycle phases using Seurat (Fig. 3e and Supplementary Fig. 7). Cluster 2 was the largest midbrain population and exhibited the highest expression of midbrain markers, with 4X cells making up 98.6% of cells in this cluster (Fig. 3b, c).

Analyzing the hindbrain cells in more detail, we classified the largest cluster as hindbrain floor plate progenitors (cluster 1), which expressed *FOXA2*, *SHH*, and *CORIN* (Fig. 3e). The hindbrain floor plate cluster comprised both H9 and 4X cells (Fig. 3b). Further examination of *HOX* gene expression within cluster 1 showed that there was an abundance of cells expressing anterior *HOXA/B* genes (Fig. 3f, and Supplementary Fig. 8). A total of 92.2% of the *HOXA/B* cells originated

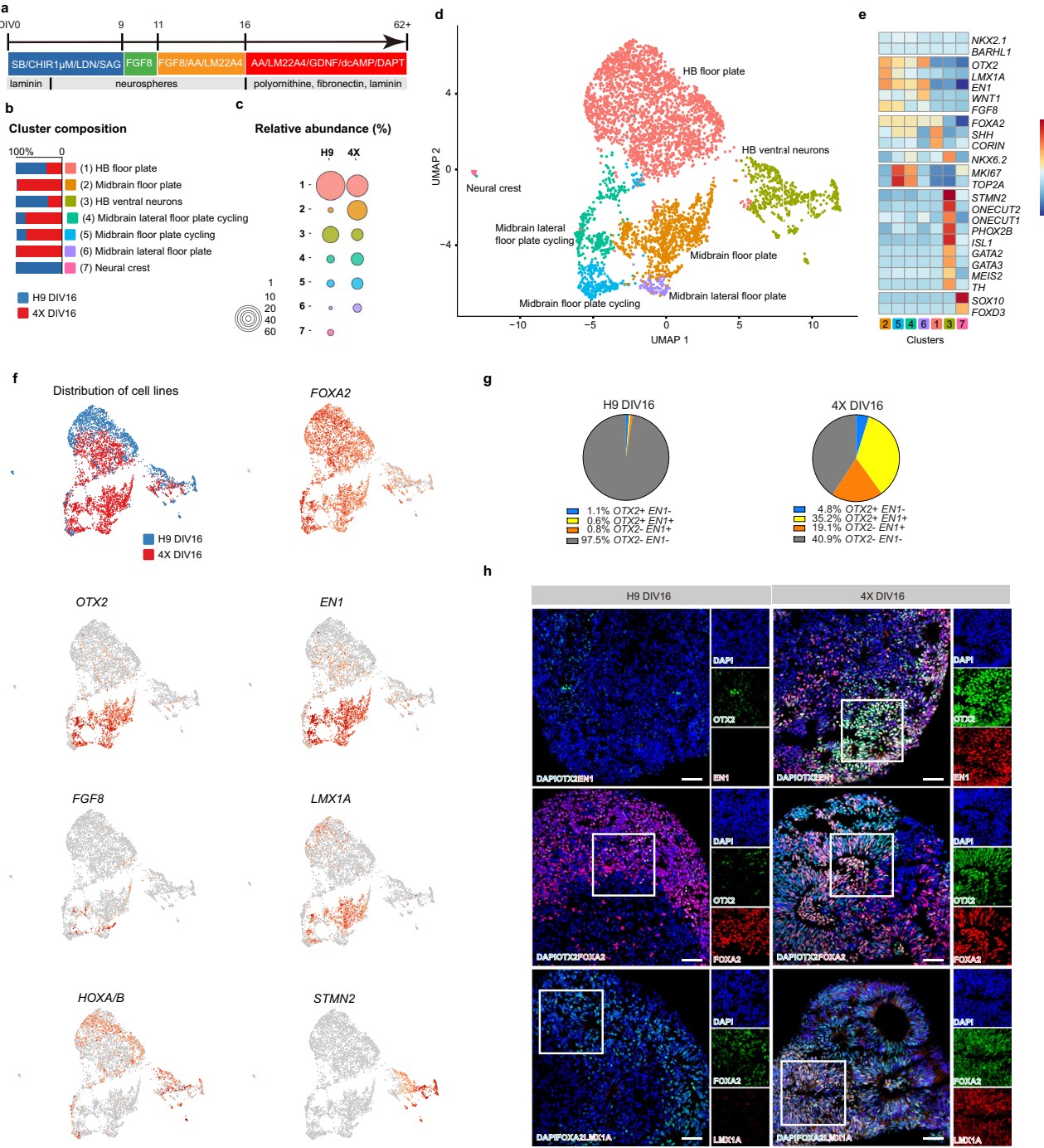

**Fig. 3 | Single-cell sequencing of H9 and 4X progenitors at DIV16 using a hindbrain differentiation protocol. a** Diagram of hindbrain differentiation protocol. **b** Graph of the cluster composition for H9 and 4X at DIV16. **c** Graph showing abundance of H9 and 4X cells in each cluster at DIV16. **d** UMAP of H9 and 4X cells at DIV16 (H9 cell spheres: n = 10; 4X cell spheres: n = 10; total of 4682 cells). **e** Heatmap of selected genes expressed in each cluster. **f** Feature plot of the contribution of each cell line to each cluster and feature plot of gene expression levels of *OTX2*, *EN1*, *LMX1A*, *FOXA2*, *FGF8*, *HOXA/B* family members and *STMN2*. **g** Percentage of H9 and 4X cells expressing *OTX2* and *EN1*. **h** Representative immunofluorescence analysis of OTX2/EN1, OTX2/FOXA2 and FOXA2/LMX1A expression in H9 and 4X cells treated with 1 μM GSK3i on DIV16, n = 3 independent experiments. DAPI was used as a nuclear stain. Scale bars, 50 μm. Source data are provided as a Source Data file.

from H9 cells, confirming that H9 cells had a more caudal identity than 4X cells (Fig. 3d, f, Supplementary Fig. 7d). We also identified a small population of early neural crest progenitors expressing *SOX10* and *FOXD3* (cluster 7), which were exclusively H9 cells (Fig. 3b–e).

By 16 DIV, a neuronal cluster (3) was present; 60% of cells in this cluster were derived from H9 cells and 40% were derived from 4X cells (Fig. 3b, d). Clusters 3 expressed high levels of *ONECUT2*, *PHOX2B*, and *ISL1*, which are markers of early-born basal-plate hindbrain motor neurons[46–48]. Additionally, we identified a small subcluster of cells from H9 cells that expressed high levels of *GATA2*, *GATA3*, and *MEIS2* but did not express *GAD1* or *GAD2* (Supplementary Fig. 7a, b, e), indicative of immature V2b GABAergic neuroblasts[47].

To support our annotation of the clusters, we autoannotated the cells using the Seurat anchoring transfer method[49]. We integrated our data with a human midbrain dataset, which we supplemented with a human vascular leptomeningeal cells (VLMC) dataset (Supplementary

Fig. 7)[50,51]. Using this approach, we identified two main ventral clusters, a midline progenitor (ProgM) cluster and floor plate medial progenitor (ProgFPM) cluster, both of which expressed *FOXA2*. In 4X cells, the ProgFPM cells expressed the midbrain markers *OTX2, LMX1A,* and *EN1,* and the ProgM cells expressed low levels of EN1, indicating a hindbrain r1 or more caudal identity. In contrast, ProgFPM and ProgM cells derived from H9 cells did not express *OTX2, LMX1A,* or *EN1* but instead expressed the hindbrain marker *HOXB2* and *HOXA/B* family members (Supplementary Fig. 7f–k). Last, for both H9 and 4X cells, a cluster of neurons from the oculomotor and trochlear nucleus (OMTN) was identified (Supplementary Fig. 7).

Immunocytochemical analysis confirmed that at 16 DIV, 4X cells produced significantly higher percentages of OTX2-positive cells (H9: 1.5% ± 1.0, 4X: 49.3% ± 8.2, $P < 0.0001$) and OTX2/EN1 double-positive cells (H9: 2.6% ± 4.4, 4X: 31.9% ± 21.7, $P < 0.005$) than H9 cells at 1 μM GSK3i (Fig. 3h and Supplementary Fig. 9a, b). Additionally, 4X cells produced significantly more LMX1A/FOXA2 double-positive cells compared to H9 cells (H9: 13.8% ± 1.9; 4X: 33.7% ± 1.1; $P < 0.05$; Supplementary Fig. 9c). Expression analysis also supported these findings (Supplementary Fig. 9d). These results demonstrate that 4X cells generate significantly more caudal midbrain cells than H9 cells when differentiated under anterior hindbrain conditions (GSK3i 1 μM).

## LR-USCs efficiently generate mesDA neurons under hindbrain conditions

To examine the extent to which 4X cells can produce mesDA neurons when differentiated using the hindbrain differentiation protocol, we extended the differentiation period to 62 DIV (Fig. 3a). We first analyzed the cells at 28 DIV and again used the Seurat anchoring method to annotate the clusters (Fig. 4a). At 28 DIV, 23.7% of the 4X cells were classified as DA neurons (clusters DA0, DA1, DA2; 18.3%, 5.2%, 0.2% respectively); these clusters all expressed *NR4A2, EN1,* and *LMX1A,* indicating that the cells were mesDA neurons (Fig. 4c, d, e, g). *TH* expression was seen in the DA1 and DA2 clusters but not in the DA0 cluster, which is in line with DA0 neurons being a population of immature DA neurons[50]. In addition to DA neurons, 4X cultures contained GABAergic, serotonergic, red nucleus, and OMTN neurons (Fig. 4e). In contrast to 4X cells, at 28 DIV, 0.5% of H9 cells were identified as DA neurons (clusters DA0 and DA1), and these clusters expressed *LMXA1* and *TH* (Fig. 4b, d, e, f); however, the H9-derived DA clusters lacked expression of *EN1,* and only cells in the DA0 cluster expressed *NR4A2.* In the H9 cell cultures, we identified OMTN neurons, serotonergic neurons, and a small cluster of VLMCs (Fig. 4d–f).

At 62 DIV, DA neurons were again detected within the 4X cultures (Fig. 4i, j, k, m). The maturation of the neurons had developed with all DA clusters expressing *TH* and maintaining the expression of mesDA markers (Fig. 4m). The total percentage of DA neurons at 62 DIV decreased to 8%, suggesting the in vitro environment was not ideal for their long-term culture. We next examined in more detail the subtypes of mesDA neurons. To distinguish between substantia nigra and ventral tegmental area (VTA) DA neurons, we assessed the expression of *GIRK2* (also known as *KCNJ6*) and CalbindinD (*CALB1*). *GIRK2* was highly expressed in all three DA clusters (DA0, DA1, DA2), whereas *CALB1* was not detected (Supplementary Fig. 10a). In H9 cultures at 62 DIV, no DA clusters were identified (Fig. 4h). The largest cluster in the H9 cell cultures was classified as a radial glial population (Rgl3; 60%), and these cells expressed known marker genes for this population (*EFNB3, NTN1,* and *SPON1*) but also *IFITM2, S100A11,* and *COL3A1,* which are markers of VLMCs (Fig. 4j–l and Supplementary Fig. 10b). Additionally, we identified a VLMC population in H9 cultures (4.5%) and only two VLMCs in 4X. The H9-derived VLMCs expressed *IFITM2, S100A11,* and high levels of *COL3A1* (Fig. 4j–l and Supplementary Fig. 10b). Interestingly, *EN1* was absent from the VLMC clusters, suggesting that it originates from a more caudal floor plate population (Fig. 4l).

To support our single-cell sequencing results, immunohistochemical analysis was performed. We used the same hindbrain differentiation protocol (1 μM GSK3i) but adapted it to generate organoids to provide an optimal environment for the survival of neurons (Supplementary Fig. 9e). At 62 DIV, the percentage of MAP2/TH double-positive neurons within organoids produced from 4X cells was significantly higher, i.e., threefold higher, than that within organoids produced from H9 cells (4X: 9.6% ± 2.1, H9: 2.8% ± 1.8, $P < 0.005$; Supplementary Fig. 9f). In the 4X organoids, we readily identified TH neurons double positive for FOXA2/TH (67.5% ± 3.0); however, in the H9 organoids, only 11.4% ± 8.4 of the TH neurons were FOXA2/TH double-positive ($P < 0.0005$; Supplementary Fig. 9g). Further examination of TH-positive neurons showed that, in accordance with our single-cell data, the most abundant population of TH neurons derived from 4X cells coexpressed GIRK2 and that there was a small population of CALB1/TH double-positive neurons (Supplementary Fig. 9h). Lastly, we examined the organoids for the presence of VLMCs. We examined the H9-derived organoids at 83 DIV and identified a population of COL3A1/COL1A1 double-positive cells with a nonneuronal morphology (Supplementary Fig. 9i); no cells positive for COL3A1 or COL1A1 were identified among the 4X cells (Supplementary Fig. 9i). Overall, our immunohistochemical analysis supported the finding from our single-cell analysis.

## DA neurons derived from LR-USCs exhibit pacemaker activity

Having shown that we can generate mesDA neurons from 4X cells under caudalizing conditions, we next wanted to examine the electrophysiological and synaptic properties of the DA neurons. We cultured the neurons on astrocytes and performed in vitro electrophysiological recordings in whole-cell patch-clamp configuration between 80 and 84 DIV. We observed that the cells developed into electrophysiologically mature neurons, as shown by their ability to generate repetitive action potentials upon somatic current injection (Fig. 5a). Recordings in current-clamp mode revealed spontaneous pacemaker activity characteristic of DA neurons, with a mix of single spikes and phasic bursts (Fig. 5b). Membrane oscillations collapsed at potentials below −50 mV. The firing frequency in our sample ranged from 1 to 5 Hz (Fig. 5c). Additionally, we differentiated 4X cells into midbrain organoids and between 65 and 80 DIV we performed electrophysiological recordings followed by biocytin backfilling and immunostaining for TH to confirm DA neuron identity (Fig. 5e, f). Of a total of 21 neurons recorded, we identified 9 that were TH positive, of which 7 responded to depolarizing current with action potentials. The activity pattern varied between tonic firing with a frequency of 1 to 3 Hz, and bursting (3 Hz, intraburst), and 4 displayed persistent spontaneous activity. Following hyperpolarizing current injection of up to 100 pA, we also identified TH neurons with a maximum sag ratio of 0.36 (at −100 pA) (Fig. 5d–f). Furthermore, HPLC analysis of cell extracts showed that the DA content in the 4X cells was significantly higher than that in the H9 cells (287.4 nmol/g in 4X cells vs. 65.2 nmol/g in H9 cells, $P = 0.002$; Fig. 5g).

We next assessed the ability of our 4X-derived mesDA neurons to establish connections with striatal medium spiny neurons (MSNs). To do so, we implemented the nigrostriatal neuronal circuit on a chip system and cocultured human iPSC-derived MSNs with mesDA neurons from H9 and 4X cells differentiated under midbrain conditions (Fig. 5h)[52]. The MSNs were cultured in a chamber with microgrooves connected to a central channel. The microgrooves were 75 μm-long allowing for dendrites to reach the central channel. On the opposing side of the channel was a chamber in which we cultured 4X or H9-derived mesDA neurons, with 500 μm-long microgrooves leading to the central channel, which only axons could traverse completely. We readily identified TH-positive axons leading to the central chamber

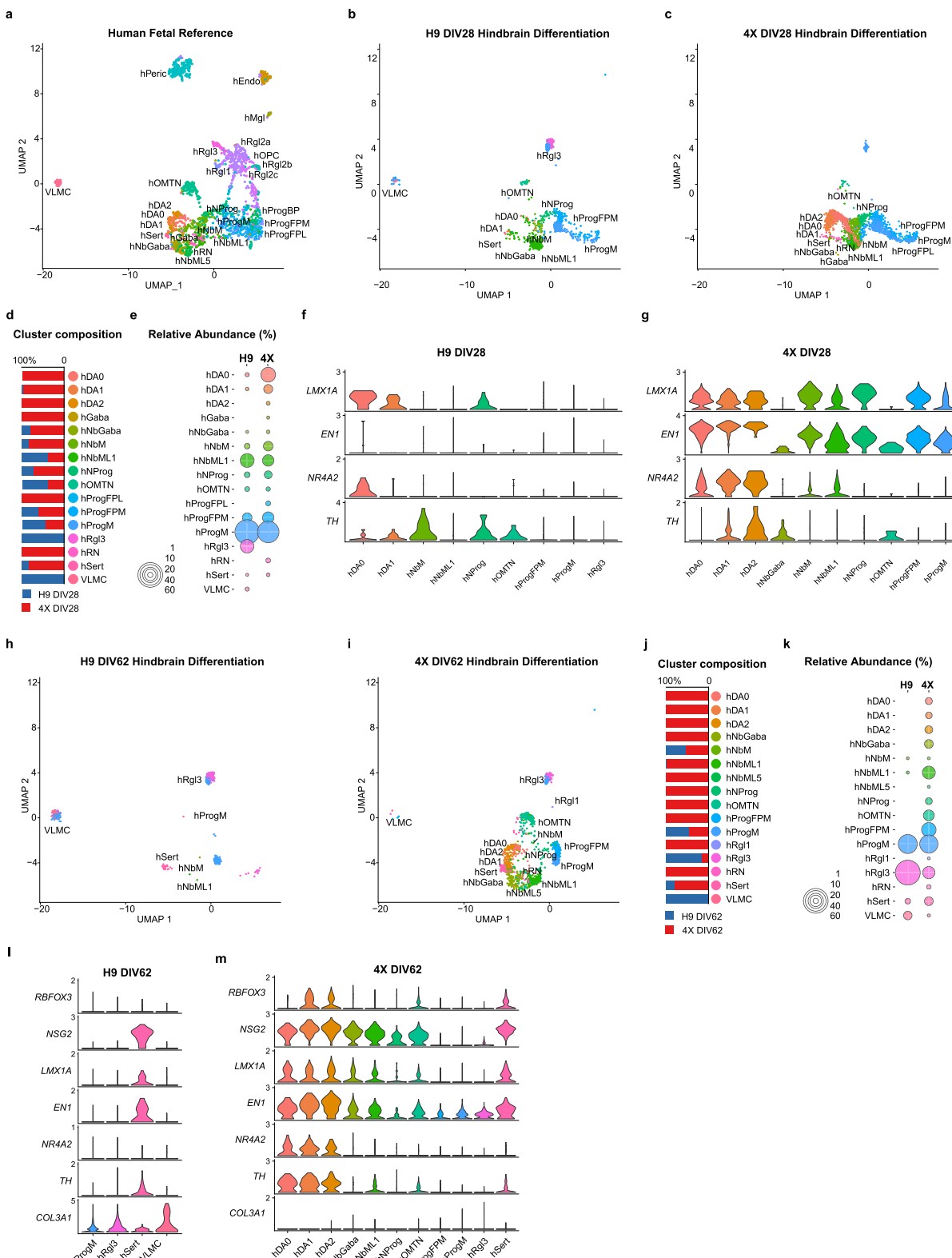

**Fig. 4 | Single-cell sequencing of hindbrain differentiated H9 and 4X cells at DIV28 and DIV62. a** UMAP of human fetal reference dataset. UMAP of H9 (**b**) and 4X (**c**) cells at DIV28 after anchoring with reference data (H9 cultures: *n* = 4, total of 2670 cells; 4X cultures: *n* = 4; total of 3280 cells). **d** Composition of cluster by H9 and 4X at DIV28. **e** Graph showing the abundance of each cluster in H9 and 4X cells at DIV28. Violin plot of H9 (**f**) and 4X (**g**) DIV28 clusters for *LMX1A, EN1, NR4A2*, and *TH*. Integration UMAP of H9 (**h**) and 4X (**i**) at DIV62. **j** Composition of the cluster by H9 and 4X at DIV62. **k** Graph showing the abundance of each cluster in H9 and 4X cells at 62 DIV (H9 cell cultures: *n* = 4, total of 3392 cells; 4X cell cultures: *n* = 4; total of 2902 cells). Violin plot of H9 (**l**) and 4X(**m**) DIV62 clusters for *RBFOX3, NSG2, LMX1A, EN1, NR4A2, TH*, and *COL3A1*. Source data are provided as a Source Data file.

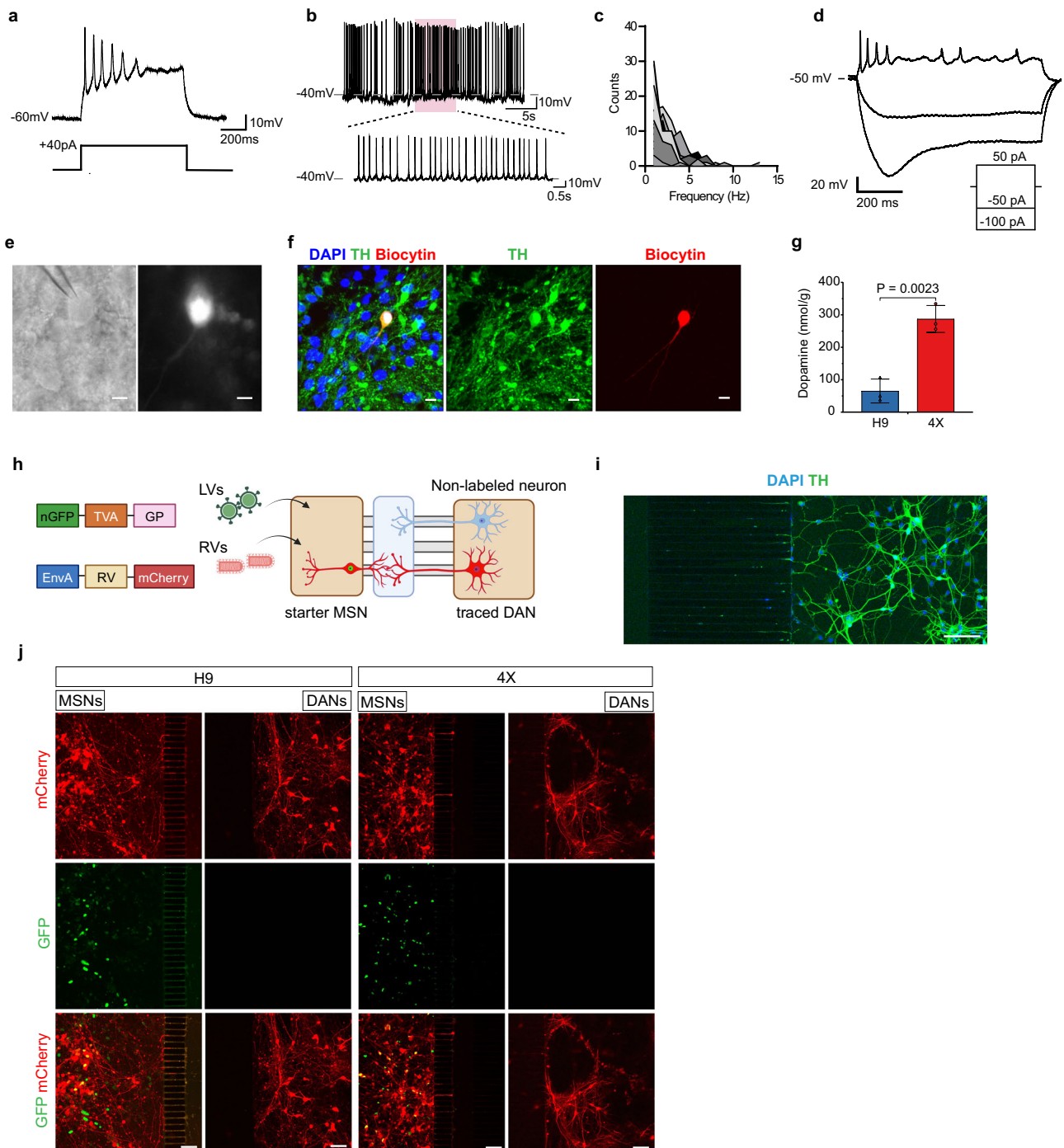

**Fig. 5 | Generation of functional ventral midbrain DA neurons in vitro.**
**a** Representative response (top trace) to a depolarizing current injection (bottom trace) showing firing of repetitive action potentials. **b** Example of spontaneous firing at a resting membrane potential of −45 mV showing burst-like events. Overshooting spikes occurred in groups interspersed by periods of subthreshold membrane oscillation. **c** Frequency distribution of spontaneous cell firing showing firing frequencies ranging between 1 and 5 Hz (*n* = 16 cells). **d** Response to a 3-step current injection protocol, displaying a sag potential upon delivery of −100 pA. **e** Phase contrast image of a patched 4X neuron and with Biocytin backfilled during whole-cell recording (*n* = 21 cells). Scale bar, 10 μm. **f** Immunofluorescence shows the expression of TH in the biocytin-labeled, recorded neuron (*n* = 9 cells). Scale bar, 10 μm. **g** Dopamine content (normalized to the protein concentration) in 4X and H9 cells at DIV79, as measured by HPLC. The data are presented as the mean ± SD; *n* = 3 biological replicates. An unpaired two-tailed t-test was used to

compare groups. **h** Illustration of the experiment with lentiviral and rabies viral vectors. Created with BioRender.com. The tracing vector transduces the MSNs, termed starter MSNs, with a nuclear GFP, the TVA receptor, and G replication factor necessary for rabies infection. After rabies viral infection, starter MSNs express mCherry, and spread the rabies viruses retrogradely to the traced neurons due to the presence of GP. **i** Representative image of TH+ mesDA neurons (DANs) seeded in the microfluidic device. *n* = 2 independent experiments. Scale bar, 50 μm. **j** Representative images of rabies traced connectivity between MSNs and DANs from H9 and 4X groups. Starter MSNs are positive for both GFP and mCherry indicating rabies viral infection, while DANs express only mCherry, indicating spreading of the rabies viruses and a stable neuronal connection to the MSNs. *n* = 2 independent experiments. Scale bars, 100 μm. Source data are provided as a Source Data file.

(Fig. 5i). We used a monosynaptic rabies virus (RV) tracing system to assess synaptic connections between MSNs and mesDA neurons. MSNs were infected with the TVA-GFP lentivirus and then with a G-protein deleted RV carrying mCherry one week later. After four days, we identified mCherry-positive (and nucGFP-negative) neurons in the DA cultures from both H9 and 4X cells, indicating that mesDA neurons derived from both PSCs and LR-USCs could form synaptic connections with MSNs (Fig. 5j).

## Analysis of hindbrain patterned 4X cells in vivo in a Parkinson's disease rat model

We next investigated how our 4X LR-USCs behave in vivo when transplanted into a rodent model of Parkinson's disease. Following our single-cell analysis, we used the same hindbrain differentiation protocol with a concentration of GSK3i (1 μM) that promotes hindbrain specification. Our choice of the hindbrain protocol was motivated by our interest to explore the potential of 4X-derived DA neurons generated under hindbrain conditions to integrate and restore function in vivo, rather than to perform a benchmarking experiment. This experimental design shares conceptual similarities with previous studies that have compared ventral midbrain patterned cells to rosette[9], or hindbrain protocols[10].

A total of 250,000 4X cells or H9 cells differentiated for 16 DIV were transplanted into the striata of nude rats with 6-OHDA-induced medial forebrain bundle (MFB) lesions 4 weeks after lesioning. A third group of lesioned rats that did not undergo transplantation was used as a lesion control (6-OHDA, see Fig. 6a for study design). For the behavioral assessment we utilized the amphetamine-induced rotation test that can reveal the imbalance in DA release between the intact and grafted striata, but due to its limitation at determining striatal reinnervation and DA subtype, we also performed the cylinder test, which is more appropriate for investigating spontaneous physiological DA release and functional integration[53,54].

At the time of transplantation, all three groups of rats exhibited a similar number of amphetamine-induced ipsilateral rotations/min (limit for inclusion: 5 rotations/min, Supplementary Fig. 11a), confirming significant loss of DA striatal innervation. All three groups showed forelimb asymmetry in the cylinder test, with the rats using mostly the ipsilateral forepaw (6-OHDA 52.8%; H9 70.9% and 4×66.3% of total) and almost never the contralateral forepaw (6-OHDA 1.5%; H9 1.2% and 4 × 0% of total) to touch the walls or land on the floor after rearing, further supporting the induction of a DA deficit by 6-OHDA (Supplementary Fig. 11b). Eight weeks posttransplantation, rats that received 4X cells showed complete correction of amphetamine-induced ipsilateral rotation (pretransplant: 10.6 vs. 8w: 0.35 rotations/min), suggesting that a sufficient amount of dopamine was released in the striatum to normalize or even overcompensate for this behavior, as suggested by the number of contralateral rotations (−3.12 rotations/min) observed at 18 weeks posttransplantation (Fig. 6b). However, H9 cell-transplanted rats presented a statistically similar number of ipsilateral rotations as the control 6-OHDA lesion group (pretransplantion: 9.8; 8w: 9.7 and 18w: 9.2 rotations/min) throughout the entire experiment, showing only a significant reduction in the number of rotations compared to pretransplantation values at 18 weeks (H9 pretransplantion: 11.5; 8w: 12.6 and 18w: 5.6 rotations/min, Fig. 6b). Analysis of spontaneous motor behavior in the cylinder test confirmed the significant improvement in 4X cell-transplanted rats, as these rats used the contralateral forelimb alone (9.6% of the time) or together with the ipsilateral forelimb (45.9% of the time) in the test at week 18 (Fig. 6c). However, both H9 cell-transplanted and 6-OHDA-lesioned rats used mostly the ipsilateral forelimb (78.3% and 75.8% of the time, respectively), used both forelimbs less than 30% of the time and almost never used the contralateral impaired forelimb when rearing in the cylinder, similar

to what was observed before transplantation (Fig. 6c). Therefore, 4X cell transplantation significantly improved both drug-induced and spontaneous motor behavior after 6-OHDA-induced lesioning of the MFB.

Postmortem histological analysis of the brains showed that rats transplanted with 4X cells had graft-derived TH-positive cells in the area of injection, i.e., the striatum, as well as in the globus pallidus, the corpus callosum and the area of the cortex above the striatum (Fig. 6d). However, H9-derived TH-positive cells remained mostly in the striatum and were also found in the globus pallidus in a few animals.

The TH-positive 4X cell graft typically extended across 6-7 coronal A-P striatal sections (in a series of 8), while the H9 cell graft occupied 4–5 sections. Indeed, the estimated TH-positive graft volume was 5 times larger in the 4X cell-transplanted rats (10.18 ± 1.3 mm³) than in the H9 cell-transplanted rats (1.96 ± 0.2 mm³) (Fig. 6g). This is in accordance with previous reports indicating that hindbrain patterned cells produce smaller grafts[10]. Quantification of graft-derived TH-positive cells (in the striatum and globus pallidus) showed that 4X cell grafts contained a slightly higher but statistically similar percentage of TH-positive cells (12.13% ± 1.7 SEM), compared to the H9 grafts (8.17% ± 2.3 SEM; $P = 0.18$) (Fig. 6h). Although the H9 and 4X grafts contained similar percentages of TH-positive cells, the significant difference in graft size and the known hindbrain composition of the H9 progenitors indicated that considerable differences in survival of the 4X and H9 progenitor cell types existed. As a result, we employed the conventional method of calculating the total number of surviving TH-positive neurons per graft for a more precise assessment of DA neuron yield. From this we identified significantly more TH-positive cells per graft in all 4X cell-transplanted rats (23,520 TH-positive cells per graft) than in the H9 cell-transplanted rats (1898 TH-positive cells per graft), resulting in a larger yield (9408 TH-positive cells per 100,000 transplanted 4X cells vs. 759 TH-positive cells per 100,000 transplanted H9 cells) (Fig. 6e, f). The increase in TH-positive cell number resulted in a significantly higher density of TH cells in the graft in the 4X cell-transplanted group (2271 ± 271 cells/mm³ vs. 988 ± 88 cells/mm³ in the H9 cell-transplanted group; $P = 0.0004$) and a significantly higher density of TH fibers extending from the graft and innervating the surrounding striatum (4X: 8.3 ± 3.6 and H9: 4.3 ± 2.4; $P = 0.017$; area covered by TH+ immunostaining; Supplementary Fig. 11c). To further assess the innervation of the TH fibers, we examined the grafts for human-specific synaptophysin (hSYP) across all grafted animals. We readily observed synaptophysin immunoreactivity along the TH fibers and on host striatal GABAergic neurons (Supplementary Fig. 11d, e), indicating that both 4X and H9-derived TH neurons could form synaptic connections with the host MSNs. Together these results are in agreement with the non-pharmacological behavioral recovery observed in the 4X cell-transplanted group.

Further examination of the grafts showed that all TH-positive neurons identified within the 4X and H9 cell grafts coexpressed the human nuclear marker human nuclear antigen (HNA) (Fig. 6i, j). In the grafts of 4X cell-transplanted rats, TH-positive neurons coexpressed FOXA2, LMX1A, and EN1, indicating that they were mesDA neurons (Fig. 6k–m). To distinguish between A9 and A10 neurons, we calculated the proportion of TH-positive neurons expressing GIRK2 and CALB1 and found that 75.4% ± 4.99 of TH-positive neurons were GIRK2-positive (Fig. 6n–p). Interestingly, we also found that TH-positive neurons derived from H9 cells were positive for FOXA2, LMX1A, and EN1 (Supplementary Fig. 11f–h) and a 70.9% ± 8.0 co-expressed GIRK2 (Supplementary Fig. 11i). This finding was in contrast to that of our 62 DIV experiments, in which TH-positive neurons derived from H9 cells rarely expressed FOXA2 (Supplementary Fig. 9g, h), suggesting that the in vivo environment is more permissive for the development and

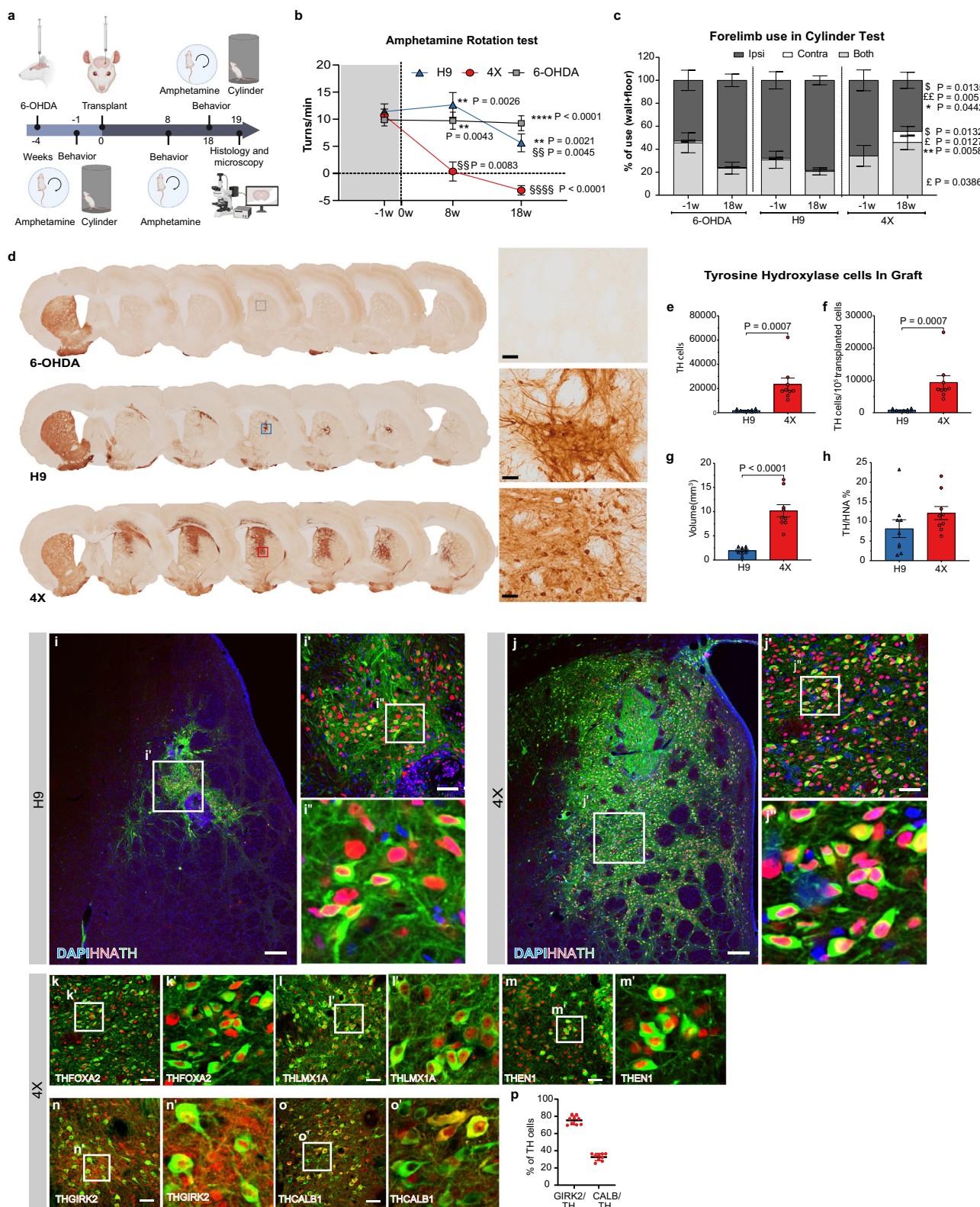

survival of TH-positive neurons than an in vitro environment. Since the in vitro data showed that H9 cells produced a large number of radial glial cells and VLMCs, we examined the expression of vascular markers in the grafts of both 4X cell-transplanted rats and H9 cell-transplanted rats. Within the H9 cell grafts, we identified a large population of COL3A1/COL1A1/HNA triple-positive cells, whereas in 4X cell grafts, we rarely detected COL1A1-positive cells coexpressing the marker HNA (Supplementary Fig. 11j, k). Additionally, we counted few 5-HT serotonergic cells across both H9 and 4x grafts. Serotonergic neurons made up 2.07% (±0.31 SEM) of the HNA-positive cells in the H9 group, and in the 4X group serotonin neurons only made up 0.54% (±0.14 SEM) of the HNA cells ($P < 0.01$; Supplementary Fig. 11l, m). Overall, the in vivo histological data showed that 4X cells could produce a robust population of mesDA neurons, consistent with the rapid motor recovery seen at 8 weeks and non-pharmacological behavioral recovery observed after 18 weeks.

**Fig. 6 | In vivo analysis of hindbrain patterned cells transplanted into a Parkinson's disease rat model. a** Overview of the in vivo study. Created with BioRender.com. **b** Amphetamine-induced rotational asymmetry. Two-way repeated measures ANOVA followed by Sidak's multiple comparison test; time: $P < 0.0001$; treatment: $P < 0.0001$. **$P < 0.01$, ****$P < 0.0001$ vs. 4X group at the same time point. §§$P < 0.01$, §§§§$P < 0.0001$ vs. the same group at week −1. **c** Cylinder test. Two-way repeated measures ANOVA followed by Sidak's multiple comparison test. Time x treatment: both: $P = 0.0259$; ipsilateral: $P = 0.0038$; contralateral: $P = 0.0244$. *$P < 0.05$,**$P < 0.01$ vs. the same group at week −1. $$P < 0.05$ vs. 6-OHDA group at the same time point. £$P < 0.05$, ££$P < 0.01$ vs. H9 group at the same time point. The data in (**b**, **c**) are mean ± SEM. 6-OHDA: $n = 7$, H9: $n = 8$, 4X: $n = 9$ rats. **d** Representative photos of coronal sections from all groups immunostained for TH. Scale bars, 50 µm. **e** Estimated numbers of TH-positive cells in the grafts. **f** Yield of TH-positive neurons per 100,000 grafted cells. **g** Volume of the TH-positive graft. **h** Estimated percentage of TH-positive cells within the HNA cells. The data in (**e**–**h**) are mean ± SEM. $n = 9$ rats per group. An unpaired two-tailed $t$ test was used to compare groups. Representative photomicrographs showing HNA-positive and TH-positive cells within H9 (**i**) and 4X cell grafts (**j**). The squares in (**i**, **j**) and (**i'-j'**) indicate the magnified areas shown in (**i'-j'**) and (**i"-j"**), respectively. Scale bars, 200 µm (**i**, **j**) and 50 µm (i'-j'). $n = 9$ rats per group. Representative immunofluorescence images of cells double-positive for TH/FOXA2 (**k**), TH/LMX1A (**l**), TH/EN1 (**m**), TH/GIRK2 (**n**) and TH/CALB1 (**o**) within 4X cell grafts. (**k'-o'**) High-power images of (**k-o**). Scale bar, 50 µm. **k–m** $n = 3$ rats, (**n**) and (**o**), $n = 9$ rats. **p** Quantitative analysis of GIRK2/TH and CALB1/TH double-positive cells within TH cells in 4X cell grafts, mean ± SD ($n = 9$ rats).

## Analysis of midbrain patterned 4X cells in vivo in a Parkinson's disease rat model

We next performed an in vivo benchmark study and transplanted H9 and 4X-derived day 16 progenitors generated under optimal midbrain conditions (Fig. 7a). Based on our quadruple FACS data, we determined that the optimal conditions for H9 cells were 0.7 µM CHIR (Fig. 2b, c, h). The optimal conditions for 4X, with its lineage restriction and the unique posterior expansion of the mesDA population, were 0.8 µM CHIR (Fig. 2b, c, h). In the first in vivo round, we obtained a large number of DA neurons in 4X cell-transplanted rats and a rapid recovery in 8 weeks; thus, we transplanted half the number of cells for both H9 and 4X (125,000 cells per rat) and extended the in vivo time to 26 weeks post-transplantation (Fig. 7a). After 18 weeks post-transplantation, 4X cell-transplanted rats showed a significant reduction in amphetamine-induced rotation compared to pretransplant and when compared to H9 cell-transplanted rats at 18 weeks ($P < 0.01$ and $P < 0.05$ respectively; Fig. 7b). However, H9 cell-transplanted rats at this time point showed no significant improvement in amphetamine-induced rotation. At 26 weeks, both 4X and H9 cell-transplanted rats showed a complete correction in rotation ($P < 0.0001$; Fig. 7b). Importantly, we performed the cylinder test, which revealed that only 4X cell-transplanted rats improved significantly in the use of the contralateral paw compared to pretransplant ($P < 0.0001$) and showed a significant improvement compared to H9 and 6-OHDA at week 26 ($P < 0.01$ and $P < 0.001$ respectively; Fig. 7c). Although H9-transplanted rats improve in the amphetamine test at 26 weeks (Fig. 7b), there was no significant increase in the use of the contralateral paw in the cylinder test (Fig. 7c). Overall, 4X cell transplantation significantly improved both drug-induced and spontaneous motor behavior after 6-OHDA-induced lesioning of the MFB.

Histological analysis of TH-positive cells within the graft revealed that the total number of TH-positive cells was significantly greater in 4X than H9 grafted rats (4X: 9,361 ± 1,225 vs. H9: 5,312 ± 506 SEM; $P = 0.0076$), resulting in a larger yield (4X: 7488 and H9: 4250 TH-positive cells per 100,000 cells transplanted; Fig. 7d, e, f), and correspondingly the TH-graft volume was significantly greater (4X: 3.32 mm³ ± 0.35 and H9: 1.85 mm³ ± 0.25 SEM, $P = 0.0056$; Fig. 7g). The percentage of TH cells per HNA cells was 14.07% (±2.65 SEM) for the H9 cell-transplanted group, whereas the 4X cell-transplanted group generated 22.86% (±3.38 SEM; $P = 0.0575$; Fig. 7h). In both H9 and 4X grafts TH-positive neurons were seen to coexpress the midbrain markers FOXA2, LMX1A, and EN1 (Fig. 7i). Additionally, the majority of TH-positive neurons coexpressed the A9 marker GIRK2 (4X: 85.19% ± 2.3, H9: 84.63% ± 3.7 SEM; Fig. 7i, j). In both H9 and 4X cell grafts, we rarely detected serotonergic neurons (between 1 and 11 5HT-positive cells detected across a series of 8). Overall, the in vivo benchmarking study showed that while both H9 and 4X cells could correct amphetamine rotation and generate dopaminergic neurons, 4X cells produced a significantly higher population of mesDA neurons and significantly improved motor function in a clinically relevant non-pharmacological behavioral test.

## Discussion

In this study, we engineered a stem cell with restricted differentiation potential, LR-USCs. By knocking out genes involved in early lineage specification, we prevented the cells from differentiating into unwanted lineages and guided their differentiation down a mesDA neuron lineage. Specifically, we examined the genes involved in the patterning of the A-P axis because of the difficulties in fine-tuning differentiation to reproducibly generate pure caudal midbrain progenitors. We targeted genes that are involved in the early specification of the hindbrain (*GBX2*) and spinal cord (*CDX1/2/4*). Importantly, the genes we targeted are not involved in the development of mesDA neurons. Our intention in targeting hindbrain and spinal cord genes was not solely to eliminate posterior cell types but, more importantly, to broaden the conditions under which mesDA neurons could be generated. As such, by knocking out these genes, we generated 4X LR-USCs, which efficiently produced caudal midbrain floor plate cells when differentiated under midbrain or hindbrain conditions.

To accurately assess the identity of mesDA progenitors, we developed a quadruple flow cytometry panel that can simultaneously analyze the expression of FOXA2, LMX1A, OTX2, and EN1. The expression of these four genes together precisely defines the caudal midbrain floor plate, and this degree of resolution has not yet been applied to mesDA differentiation protocols. Using a midbrain differentiation protocol, recently approved in a human clinical trial, we showed that 69% of 4X progenitors expressed these four markers compared to 25% of PSC-derived progenitors. Importantly, we demonstrated that 4X lineage-restriction ensured OTX2/EN1 (caudal midbrain populations) are more abundant than more anterior populations and r1 hindbrain cell types compared to control lines. Furthermore, we used EN1/TH to distinguish between mesDA neurons and non-mesDA neurons and show that 4X LR-USCs produce significantly more than their hPSC controls. Notably, a protocol which incorporates a neuroepithelial manual isolation step reports high percentages of OTX2/EN1 double-positive progenitors and produces a high percentage of EN1/TH double-positive DA neurons after extended differentiation in vitro[11,21]. Similarly, hPSCs carrying a LMX1A/EN1 dual-reporter show progenitors sorted for LMX1A/EN1 yield a high percentage of TH-positive neurons[55]. These results are consistent with ours that demonstrate that successful specification to a caudal midbrain floor plate is essential for generating mesDA neurons.

During development, when GRNs are disrupted, cells can adopt alternate fates even though they are spatially located in regions where extrinsic signals are tuned to generate other lineages. We investigate how our 4X cells – which have lost key transcriptional determinates of hindbrain regions – would respond when differentiated in hindbrain conditions. The majority of H9 cells adopted a hindbrain floor plate identity while a significant proportion of 4X cells did not and instead generated a caudal midbrain OTX2/EN1-positive progenitor identity; indicating that despite the in vitro environment 4X cells could adopt an alternate fate similar to what is observed with developmental knockout studies. The increase in caudal midbrain progenitors also

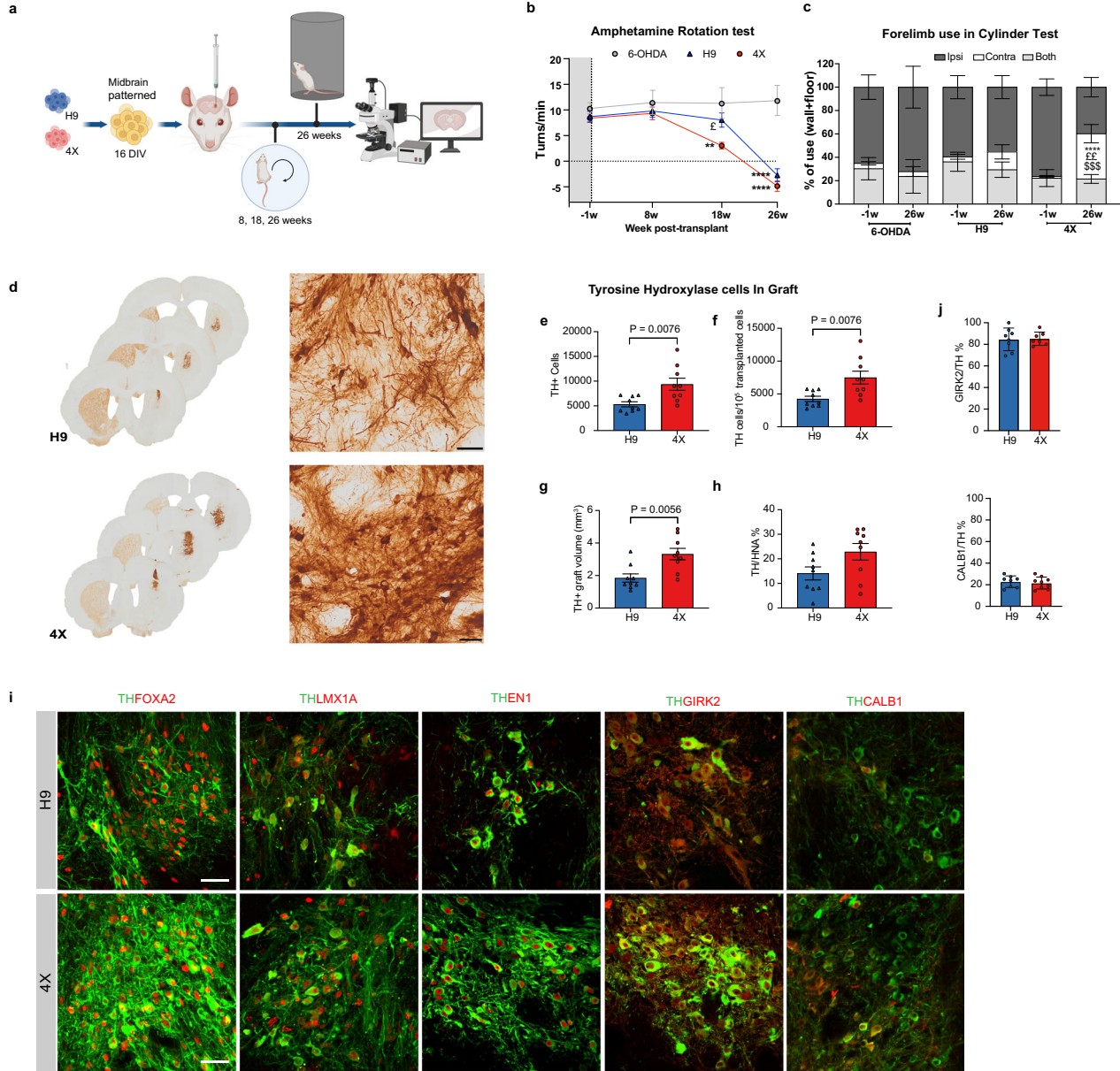

**Fig. 7 | In vivo analysis of midbrain patterned cells transplanted into a Parkinson's disease rat model. a** Overview of the in vivo study. Created with BioRender.com. **b** Amphetamine-induced rotational asymmetry. Two-way repeated measures ANOVA followed by Sidak's multiple comparison test; time: $P < 0.0001$; treatment: $P = 0.0009$. **$P = 0.0072$ and ****$P < 0.0001$ vs. the same group at week −1. £$P = 0.0225$ vs. H9 group at the same time point. The data are presented as the mean ± SEM. 6-OHDA group $n = 6$, H9 group $n = 9$, and 4X group $n = 9$ rats. **c** Cylinder test. Two-way repeated measures ANOVA followed by Sidak's multiple comparison test. Time x treatment: contra: $P = 0.0039$; ****$P < 0.0001$ vs. the same group at week −1. $$$$P = 0.0003$ vs. the 6-OHDA group at the same time point. £ £$P = 0.0048$ vs. the H9 group at the same time point. The data are presented as the mean ± SEM. 6-OHDA group $n = 4$, H9 group $n = 6$, and 4X group $n = 7$ rats.

**d** Representative photos of coronal sections H9 and 4X cell-transplanted groups for TH. Scale bars, 50 μm. **e** Estimated numbers of TH-positive cells in the grafts. **f** The yield of TH-positive neurons per 100,000 grafted cells. **g** TH-positive graft volume. **h** The estimated percentage of TH-positive cells within the HNA cells. The data in e-h are mean ± SEM. $n = 9$ rats per group. An unpaired two-tailed $t$ test was used to compare groups in (**e**, **f**, **h**) and a Mann−Whitney two-tailed test for (**g**). **i** Representative immunofluorescence images of cells double-positive for TH/FOXA2, TH/LMX1A, TH/EN1 ($n = 3$ rats), TH/GIRK2 and TH/CALB1 ($n = 9$ rats) within H9 and 4X cell grafts. Scale bars, 50 μm. **j** Quantitative analysis of the immunofluorescence data showing the percentages of GIRK2/TH and CALB1/TH double-positive cells within TH cells in H9 and 4X cell grafts. The data are mean percentage ± SD (CALB1, H9: $n = 8$, 4X: $n = 9$; GIRK2, H9: $n = 8$, 4X: $n = 7$ rats).

correlated with the generation of mesDA neurons that were functional and displayed characteristic pacemaker activity.

When hindbrain patterned 4X cells were transplanted in vivo into rats with 6-OHDA-induced MFB lesions, motor behavior improved, with amphetamine-induced rotation being fully corrected at only 8 weeks post-transplantation. Indeed, histological examination of 4X cell grafts showed an estimated number of 23,520 TH-positive cells after 250,000 cells were transplanted. This large number of TH-positive cells support the observed rapid behavioral recovery. Furthermore, we showed that the TH-positive cells expressed important mesDA neuronal markers (FOXA2, LMX1A, and EN1), and the majority of TH-positive cells were GIRK2-positive, demonstrating that there was an abundance of putative A9 DA substantia nigra neurons. Interestingly, hindbrain patterned H9 cells generated small grafts, which has previously been reported[10]; however, the percentage of DA neurons in the H9 grafts were similar to 4X, which was in contrast to what we saw

in vitro. These results indicate that despite the hindbrain specification the few DA neurons in the H9 grafts preferentially survived in vivo, suggests that the denervated striatum is more permissive for the development and survival of TH-positive neurons than in vitro[56].

Amphetamine forces the release of DA from the grafted cells in a mechanism independent of the action-potential-induced vesicular release[57]. Consequently, the amphetamine induced-rotation test does not directly measure graft integration as the spillover of extracellular DA can act on proximal and distal post-synaptic targets. Thus, in this study, we included the cylinder test, a non-pharmacological behavioral test, to carefully assess the functional integration of the graft. We observed spontaneous use of the affected forelimb in 4X-transplanted rats in the hindbrain patterned in vivo experiment, whereas H9-transplanted rats did not improve. In our in vivo midbrain patterned experiment, H9-transplanted rats contained graft-derived mesDA neurons and showed recovery in the amphetamine test, though in insufficient numbers to generate a function recovery in the cylinder test. However, with the same in vivo maturation time, 4X-transplanted rats contained a significantly higher number of DA neurons and produced a significant recovery in the cylinder test. Our results underscore the importance of using a clinically relevant non-pharmacological behavioral test when examining the functional integration and if adequate numbers of A9 DA neurons are present. A report of a patient with few grafted DA neurons has revealed that over time – as the disease progressed – the efficacy of the graft declined, highlighting the need for successful grafts of high purity and with a clinically relevant number of neurons[58]. In this study, we demonstrate that lineage restriction can yield higher purity and cell number, which are vital not only for initial motor improvements in patents but may also benefit the long-term success of the graft.

Thus far, we have described how the knockout of four selected genes can dramatically increase the specification of PSCs to mesDA progenitors and neurons by restricting the cell types along the A-P axis that they can differentiate into. It is possible to further restrict cell fate by knocking out additional genes to prevent differentiation into remaining populations of unwanted cells, which would further enhance the generation of mesDA neurons. Specifically, our single-cell sequencing data showed that 4X cells are capable of producing neighboring neuronal populations not restricted by the knockouts, and we speculate that by targeting their unique transcriptional determinates they could be eliminated. Furthermore, in this study we did not examine if the same improvements are seen with other protocols, such as the CHIR-boost protocol, or if a reduced set of genes can be knocked out when differentiating the cells with current midbrain protocols. Adding and refining the set of genes knocked out in combination with a midbrain protocol may produce a highly pure population of mesDA neurons. Overall, the present LR-USC line prominently highlights important characteristics of this approach; specifically, the ability to restrict undesired lineages and generate high proportions of mesDA precursors and under a broader range of growth factor conditions compared to unedited lines. This has significant advantages for clinical applications, allowing for easier upscaling and reproducibility, and reduced cell line variability. Lineage restriction can also be used to produce other cell types. By deleting different sets of genes, LR-USCs can be designed to preferentially generate other neural populations or cell types from other germ layers, which can be used for cell transplantation therapy or drug discovery for the treatment of a range of disorders.

## Methods

### hPSC culture
hESCs H9 (WA09; WiCell) and H1 (WA01; WiCell) and iPSCs GBA-002-C3[45] (DANi-002C) were maintained on irradiated human fibroblasts in KSR medium consisting of DMEM/nutrient mixture F-12 supplemented with 1% nonessential amino acids (NEAAs), 2 mM glutamine, 0.1 mM β-mercaptoethanol, 0.5% pen/strep and 20% knockout serum replacement. The KSR medium was supplemented with FGF2 (15 ng/ml; Peprotech) and Activin A (15 ng/ml; R&D Systems). Every seven days, the cells were manually passaged, and fragments were transferred to a freshly prepared gelatin-coated dish containing irradiated fibroblasts[59].

### Differentiation into CNPs
hESCs were differentiated into CNPs as described previously[37]. Briefly, hESC fragments were cut from colonies growing on irradiated feeders (CCD-1079Sk, ATCC) and plated in vitronectin-coated plates in N2B27 medium containing neurobasal medium (NBM) and DMEM/F-12 supplemented with 1% N2 supplement at a 1:1 ratio, 1% B27 supplement minus vitamin A, 1% insulin/transferrin/selenium-A (ITS- A), 0.3% glucose, 1% Glutamax supplement, and 0.5% penicillin/streptomycin (all from Life Technologies). The medium was supplemented with SB431542 (SB; 10 µM, Tocris Bioscience) and CHIR99021 (CHIR; 3 µM, Stemgent) for 4 days. For the 11 and 32 DIV CNP differentiation protocol, cells were cultured as described above and supplemented with SAG (400 nM, Merck Millipore). After day 4 the colonies were dissected into 0.5 mm pieces and cultured in suspension in low-attachment 96-well plates (Corning) in N2B27 medium supplemented with FGF2 (20 ng/ml; PeproTech) and SAG (400 nM) until day 11. From day 11 to 16 N2B27 medium was supplemented with FGF2 (20 ng/ml; PeproTech). From day 16 to 32, cells were cultured in B27 base media containing NBM with 2% B27 without Vitamin A, 1% Glutamax supplement, and 0.2% penicillin/streptomycin (all from Life Technologies) and supplemented with ascorbic acid (200 µM, Sigma-Aldrich), LM22A4 (2 µM, Tocris), GDNF (10 ng/mL), dcAMP (500 µM, Sigma-Aldrich), DAPT (1 µM, Tocris).

### mesDA neuron differentiation
hPSC cells were seeded at a density of 10,000–18,000 cells/cm² on Laminin (L2020, 10 µg/ml in PBS + +, Sigma-Aldrich) pre-coated plates in N2 basal media containing NBM and DMEM/F-12 supplemented with 1% N2 supplement at a 1:1 ratio, 0.5% Glutamax supplement, and 0.2% penicillin/streptomycin (all from Life Technologies). The medium was supplemented with SB (10 µM), CHIR (0.6–0.8 µM), LDN-193189 (LDN; 100 nM, Stemgent), SHH C25II (SHH, 500 ng/mL, R&D systems), and SAG (400 nM) for 9 days. On day 9 to day 11, the media was changed to be N2 basal media with FGF8b (100 ng/mL, Peprotech). On day 11, cells were treated with Accutase and replated at 800,000 cells/cm² on Laminin coated plates and cultured in B27 base media containing NBM with 2% B27 without Vitamin A, 1% Glutamax supplement, and 0.2% penicillin/streptomycin (all from Life Technologies). The medium was supplemented with FGF8b (100 ng/mL), ascorbic acid (200 µM, Sigma-Aldrich), LM22A4 (2 µM, Tocris), BDNF (20 ng/mL, R&D systems), dcAMP (500 µM, Sigma-Aldrich), GDNF (20 ng/mL, R&D systems) and DAPT (10 µM, Tocris) and cells were cultured until day 16 with media changed every day. On day 16, cells were either collected for analysis or dissociated with Accutase and replated at 600,000–800,000 cells/cm² on poly-L-ornithine (0.005%) and LN521 (10 µg/mL) coated plates in B27 basal media supplemented with ascorbic acid (200 µM), LM22A4 (2 µM), BDNF (20 ng/mL), dcAMP (500 µM), GDNF (20 ng/mL) and DAPT (10 µM) and cultured until day 30. Media was changed every second day. Y-27632 (10 µM, Tocris) was added to the culture media on day 0, day 11, day 16 and day 24.

### Hindbrain differentiation
Neurons were generated by implementing previously described protocols with minor modifications[16]. Briefly, from day 0 to day 9, cells were grown in N2B27 medium supplemented with SB (10 µM), CHIR (1 µM), LDN (100 nM), and SAG (400 nM). On day 4, the colonies were cut into fragments and cultured in suspension. From day 9 to 11, the supplements in the medium were replaced with FGF8 (100 ng/ml).

From day 11, the medium was supplemented with FGF8 (100 ng/ml), LM22A4 (2 μM), and ascorbic acid (200 μM). On day 16, the cells were dissociated with Accutase and subsequently grown on culture plates coated with polyornithine, fibronectin, and laminin (all from Sigma-Aldrich). Neural differentiation medium containing of 1% B27 supplement, 0.5% penicillin/streptomycin, and 0.5% Glutamax was supplemented with ascorbic acid (200 μM), LM22A4 (2 μM), DAPT (1 μM), GDNF (10 ng/ml), and dcAMP (500 μM). The medium was changed every second day until the end of the experiment. Alternatively, on day 16, the cultured cells were maintained in suspension to generate organoids.

### Generation of CRISPR vectors

A pLV-4gRNA-GBX2-RFP (Addgene #192288) lentiviral plasmid containing four CRISPR target sites in GBX2 was generated using the multiplex CRISPR lentiviral vector system[60]. First, oligos containing the 20 bp protospacer sequence against the four CRISPR target regions in *GBX2* (Supplementary Table 1) were cloned by BbsI digestion and ligated into the following entry plasmids, i.e., ph7SK-gRNA, phU6-gRNA, pmU6-gRNA and phH1-gRNA (Addgene #53189, 53188, 53187 and 53186), to generate four gRNA GBX2 entry plasmids. Second, using the Golden Gate recombination method, the pLV-GG-hUbC-dsRED plasmid (Addgene #84034) and the gRNA GBX2 entry plasmids were recombined by BsmBI digestion and ligation to form the final pLV-4gRNA-GBX2-RFP plasmid. A multiplex CRISPR lentiviral plasmid containing CRISPR targets in the *GBX2* and *CDX1/2/4* genes was generated in a similar manner as above. Four entry plasmids, i.e., phH1-GBX2-gRNA, phU6-CDX4-gRNA, pmU6-CDX1-gRNA, and ph7SK-CDX2-gRNA (Addgene #192510, 192511, 192512, 192513), were generated and recombined with pLV-hUbC-Cas9-T2A-GFP (Addgene #53190), resulting in the generation of the pLV-hUbC-GBX2-CDX124-Cas9-T2A-GFP plasmid (Addgene #192287; see Supplementary Table 1 for gRNA sequences). The pBS-GBX2-CDX124-Ef1a-EM7-mCherry-Neo plasmid (Addgene #192823) was generated by Golden Gate recombination of phH1-GBX2-gRNA, phU6-CDX4-gRNA, pmU6-CDX1-gRNA, ph7SK-CDX2-gRNA and pBS-BsmbI-LoxP-Ef1a-EM7-mCherry-Neo-PolyA-LoxP (Addgene #192822) in a similar manner as above.

### Generation of knockout cell lines

Three lentiviral plasmids, pLV-4gRNA-GBX2-RFP, pLV-hUbC-GBX2-CDX124-Cas9-T2A-GFP, and lentiCas9-Blast (Addgene # 52962), were used to produce lentiviruses. Lentiviral production was performed as described previously[61]. To generate the GBX2 knockout cell line, H9 cells were transduced with LV-4gRNA-GBX2-RFP and lentiCas9-Blast, and after three days, transduced cells were selected using 10 μg/ml blasticidin for 6 days (Supplementary Fig. 1). FACS was then used to separate single RFP-positive cells in a 96-well plate using a 561 nm laser on a FACSAriaIII (BD Biosciences, San Jose, CA). Indels at the corresponding target sites in the clones were analyzed by genomic PCR. To generate the 4X knockout cell line, H9 cells were infected with LV-hUbC-GBX2-CDX124-Cas9-T2A-GFP, and after 7 days, single GFP-positive cells were sorted by FACS (Supplementary Fig. 2). Allele-specific mutations in both the *GBX2*-/- and 4X cell lines were confirmed using whole-exome sequencing. Whole-exome sequencing and mapping were performed by BGI (BGI, Copenhagen). Integrated Genome Browser V 2.10.0 was used to identify allele-specific mutations. To identify large deletions that could not be mapped by the alignment tools, individual sequencing reads were extracted from the FastQ files using Grep and manually analyzed. To generate 4X-GBA-C7 and 4X-GBA-C8, DANi002C iPSCs were transduced and sorted by FACS using the same method lentiviral method as 4X. The clones of 4X-GBA-C7 and 4X-GBA-C8 were analyzed for indel by PCR amplification of genomic regions and Illumina-based sequencing of amplicons was performed by Alzenta (Alzenta Life Sciences, Leipzig, Germany; Supplementary Fig. 5). Sequences with a frequency of less than 5% were

filtered out. To generate 4X-H1-NC1, H1 hESCs were transfected using the P3 Primary Cell 4D-Nucleofector™ X kit (Lonza) with a Lonza 4-D Nucleofector (program: CB-150). A CRISPR-Cas9 ribonucleoprotein complex (5 μg; Integrated DNA Technologies) containing equal amounts of sgRNA against *CDX1, CDX2, CDX4*, and *GBX2*; Supplementary Table 1 and 250 ng of the plasmids pBS-GBX2-CDX124-Ef1a-EM7-mCherry-Neo (Addgene #192823) and pCAG-SpCas9-2A-GFP-noITR (Addgene #118415) were included in the transfection solution. After two days, FACS was used to separate single RFP-positive cells in a 96-well plate using a 561 nm laser on a FACSAriaIII (BD Biosciences, San Jose, CA). Clones were analyzed by PCR amplification of genomic regions and Illumina-based sequencing of amplicons was performed by Alzenta (Supplementary Fig. 5).

### QPCR and NanoString

For QPCR and NanoString experiments, RNA was extracted using a Qiagen RNeasy Mini Kit and treated with DNase I according to a standard protocol. cDNA was generated from 500 ng of total RNA using Superscript III and random primers following the manufacturer's instructions. For QPCR, TaqMan Universal Master Mix II without UNG and TaqMan probes were used (Supplementary Table 2). NanoString experiments were performed using the NanoString nCounter SPRINT (NanoString Technologies) according to the manufacturer's instructions. Briefly, 200 ng of total RNA was used, and reporter probes were hybridized for 20 h at 65 °C. A custom designed NanoString CodeSet consisting of a panel of capture and reporter probes designed to target 100 nucleotides of the gene of interest and a panel of housekeeping genes was used (Supplementary Table 3). RNA expression data were normalized to the expression of housekeeping genes.

### Immunofluorescence

Cells cultured on glass coverslips or suspended in culture plates as spheroids were collected. The samples were washed with PBS two times, fixed in 4% paraformaldehyde (PFA) in PBS at 4 °C for 15 min (glass coverslips) or 2 h (spheroids), and washed 3 times with PBS for 10 min each. The spheroids were transferred to 20% sucrose in PBS, incubated at 4 °C overnight and embedded in OCT (Tissue-Tek). Sections were cut at a thickness of 10 μm using a cryostat (Crostar NX70) at −20 °C. The coverslips and sections were incubated in 0.25% Triton X in PBS (PBST) for 10 min and blocked in 5% donkey serum (Almeco) in PBST for 1 h at room temperature. The following primary antibodies were applied overnight at 4 °C: goat anti-OTX2 (1:500, R&D Systems, cat# AF1979), mouse anti-CDX2 (1:200, BioGenex, cat# MU392-UC), mouse anti-Engrailed1 (EN1, 1:40, DSHB, cat# 4G11-s), rabbit anti-EN1 (1:50, Merck, cat# HPA073141), rabbit anti-FOXA2 (1:500, Cell Signaling, cat# 8186), goat anti-FOXA2 (1:200, R&D Systems, cat# AF2400), rabbit anti-LMX1A (1:5000, Millipore, cat# AB10533), mouse anti-TH (1:2000, Millipore, cat# MAB318), rabbit anti-TH (1:1000, Pel Freez, cat# P40101-150), chicken anti-MAP2 (1:2500, Abcam, ab92434), rabbit anti-GIRK2 (1:500, Alomone, cat# APC-006), mouse anti-CALB1 (1:5000, SWANT, cat #300), rabbit anti-Collagen3A1 (1:1000, NovusBio, cat# NB120-6580), sheep anti-hCOL1A1 (1:200, R&D Systems, cat# AF6220), and mouse anti-HNA (1:200, Abcam, cat# ab191181). Biocytin backfilling organoid slices were stained with mouse anti-TH and streptavidin conjugated with Alexa Fluor™ 568 (1:1000, Invitrogen). After the cells were washed with PBST three times for 10 min each, corresponding secondary antibodies (1:200, Jackson ImmunoResearch Laboratories or 1:1000, Invitrogen) were applied for one hour at room temperature. After the secondary antibodies were removed, the cells were washed three times with PBST for 10 min each in the dark. The nuclei were counterstained with DAPI (1 μg/ml, Sigma-Aldrich) and rinsed with PBS three times for 5 min each. The slides or coverslips were mounted with PVA-DABCO. Images were captured with a confocal microscope (Zeiss LSM 780 and LSM 800) and Zen software.

## Intracellular flow cytometry analysis

Cells cryopreserved at 16 DIV were thawed, and dead cells were labeled with a fixable near-infrared viability dye (1:1000, Invitrogen) in neurobasal medium (Gibco) supplemented with 1% N2 supplement (Gibco) for 15 min at room temperature protected from light. The cells were then washed once with FACS buffer (1% bovine serum albumin in PBS-/-), pelleted by centrifugation at 400 g for 10 min, and fixed and permeabilized using the Transcription Factor Buffer Set (BD Biosciences) according to the manufacturer's instructions. For each sample, $0.5 \times 10^6$ fixed cells were incubated with a cocktail of primary antibodies against FOXA2 (1:300, PE-conjugated, Miltenyi Biotec), OTX2 (1:300, VioB515-conjugated, Miltenyi Biotec), EN1 (1:50, Atlas Antibodies), and LMX1A (1:2500, Novo Nordisk) in Perm/Wash buffer (BD Biosciences) for 30 min at 4 °C protected from light. The cells were then washed three times with Perm/Wash buffer, pelleted by centrifugation at 800 g for 3 min, and incubated with a cocktail of secondary antibodies against mouse IgG (1:4000, Alexa Fluor 647-conjugated, Invitrogen) and rabbit IgG (1:200, BV421-conjugated, Jackson Immuno Research) for 30 minutes at 4 °C protected from light. Finally, the cells were washed twice with Perm/Wash buffer, pelleted by centrifugation at 800 g for 3 min, resuspended in 200 μL of FACS buffer, and passed through a 40 μm cell strainer (Merck Millipore). The samples were acquired using a four-laser CytoFLEX S flow cytometer (Beckman Coulter, Indianapolis, IN, USA) with CytExpert software, and 20,000 live cell events were recorded per sample. Compensation was performed using compensation beads (anti-REA Compensation Beads, Miltenyi Biotec; UltraComp eBeads™ Compensation Beads, Invitrogen; ArC Amine Reactive Compensation Kit, Invitrogen). FCS files were exported and analyzed using FlowJo software (v10.7.2, Ashland, OR, USA). Debris, doublets, and dead cells were filtered out. Gates were set based on appropriate unstained, secondary antibody only, and fluorescence-minus-one (FMO) control samples and verified using biological positive and negative control samples (hESC-derived ventral midbrain and spinal cord neural stem cells, respectively). Prior to antibody staining, analysis of unstained live and fixed cells confirmed that the GFP signal in 4X gene-edited cells was quenched during the fixation/permeabilization procedure, allowing the use of a VioB515-conjugated antibody.

## Quantification of immunofluorescence images

The immunofluorescence images were quantified using StarDist nuclei detection and semiautomatic object-based colocalization analysis in ImageJ software (1.53)[62,63]. The Colocalization Image Creator Plugin was used to process the multichannel immunofluorescence images into multichannel binary and grayscale output images. Binary output images were generated by processing input channels for ImageJ filters that applied an automatic local intensity threshold, radius outlier removal, watershed segmentation, eroding, hole filling, Gaussian blurring and maximum algorithms. Binary objects of an inappropriately small size were further removed from the output images via a defined minimum area size. To improve the visualization of the colocalization signals, the object overlap was restricted to the nuclei of the cells. The accuracy of the binary object segmentation was visually verified via grayscale output images. Once verified, the binary objects, representing either individually labeled or colabeled cells, were quantified automatically using the Colocalization Object Counter plugin. All immunofluorescence images were analyzed with conserved binary object segmentation settings. A minimum of 3 random fields from a minimum of 3 independent experiments captured at 20x, 40x, and 63x were used for the quantification of cells at 16 DIV, positive for OTX2, OTX2/EN1, FOXA2/LMX1A, and at 30 DIV positive for TH, EN1/TH, FOXA2/TH. The percentage of MAP2/TH, FOXA2/TH double-positive cells at 62 DIV differentiated under hindbrain condition were assessed by analysing tile-scan images of immunostained organoid cryosections. Quantification of GIRK2/TH double-positive and CALB1/

TH double-positive cells within the graft was performed blindly by analyzing 4 nonoverlapping images taken at 20x from 2 sections per graft per animal.

## RNA sequencing and data analysis

Library construction, sequencing and initial data filtering, including adapter removal, were performed by the BGI Europe Genome Center. Total RNA was subjected to oligo dT-based mRNA enrichment. Sequencing of 100 bp paired-end reads was performed on the DNBseq platform. More than 20 million clean reads were obtained per sample. The reads were aligned to the Human genome build hg38 (Ensemble release 92) using HISAT2 aligner (v2.1.0)[64]. Transcript quantification was performed using FeatureCount (v1.6.4), and the read counts were normalized for effective gene length and sequencing depth to yield transcripts per kilobase million (TPM)[65]. Differentially expressed genes were identified from count tables using edgeR (v3.32)[66]. Centering and univ variance scaling were applied to TPM values to construct heatmaps and perform principal component analysis (PCA) by Clustvis using SVD with imputation[67].

## Single-cell RNA-seq and data analysis

On 16, 28, and 62 DIV, cultured cells were dissociated into single cells using Accutase. On 16 DIV, neurospheres ($n = 10$ biological replicates per cell line) were pooled together, on 28 DIV, four biological replicates per cell line were pooled together, and on 62 DIV, four biological replicates per cell line were pooled together. To construct the library, the 10X Genomics Chromium Next GEM Single Cell 3' kit v 3.1 was used according to a standard protocol. Each of the six groups (H9 DIV16, 4X DIV16, H9 DIV28, 4X DIV28, H9 DIV62, and 4X DIV62) was run in separate lanes of the Chromium controller, and a total of 8000 cells were loaded per lane. Next-generation sequencing was performed by the NGS Core Center, Department of Molecular Medicine, Aarhus University Hospital, Denmark. Sequencing was performed on an Illumina NovaSeq instrument. The Cell Ranger Single-Cell Software Suite (v 3.1.0) was used for sample demultiplexing, barcode processing, and single-cell 3' gene counting. The reads were aligned to the human GRCh38 reference genome. Further analysis, including quality filtering, dimensionality reduction, and application of standard unsupervised clustering algorithms, was performed using the Seurat R package (v 3.2.1). To exclude outlier cells, the number of genes expressed in each cell was plotted for each sample to select the optimal allowed minimum number of genes per cell. The minimum numbers of genes per cell were set to 3000 for H9 DIV16, 2000 for 4X DIV16, 2500 for H9 DIV28, 2500 for 4X DIV28, 3000 for H9 DIV62, and 3000 for 4X DIV62. Cells with a high percentage of reads mapped to mitochondrial genes were also removed. For DIV16 and DIV28 samples, all cells with more than 10% mitochondrial RNA were removed; for DIV62, the limit was 15%. The R package DoubletFinder (v.2.0.3) was used to remove cell doublets from the single-cell transcriptome data, with the expected percentage of doublet cells being set at 7.5%. The single-cell data were normalized by dividing the gene counts of each cell by the total counts for that cell, multiplying by a scaling factor of 10,000, and natural-log transforming the result. Dimensionality reduction was performed using the UMAP technique. Clustering was performed by Seurat's graph-based clustering approach using the FindClusters function, with the resolution set to 0.6. Various single-cell plots were generated using Seurat in R.

## Single-cell reference annotation method

For the annotation of 10X Genomics single cells data, a reference annotation dataset was established. This reference consisted of the fetal human ventral midbrain single-cell data merged with VLMC cell type data extracted from Human motor cortex single-cell data[50,51]. Using Seurat (v 4.1.1), the reference annotation dataset was normalized and scaled, variable features detected, followed by Principal

Component Analysis (PCA) and UMAP dimensional reduction. Each of our 10X Genomics single cell samples (query) were annotated by projection onto the reference UMAP structure. This was done by finding anchors between query and reference [Seurat::FindTransfer-Anchors] and using these anchors to map [Seurat::MapQuery] the query cells to the reference UMAP[49]. This annotated the query cells with predicted cell type from the reference dataset.

### Electrophysiology

Electrophysiological recordings of 4X cells were performed at 80–84 DIV. Then, 4X cells cocultured with astrocytes on 13 mm Ø coverslips were transferred to a recording chamber following a progressive transition from culture medium to artificial cerebrospinal fluid (aCSF) by adding five drops (200 μL each) of aCSF to the cultured medium over 20 s. After being transferred into the recording chamber, the coverslips were continuously perfused at room temperature with aCSF containing (in mM) 119 NaCl, 2.5 KCl, 26 NaHCO₃, 1 NaH2PO₄, 11 D-glucose, 2 CaCl₂, and 2 MgCl₂ (adjusted to pH 7.4).

The recording chamber was mounted on an upright microscope (Scientifica) linked to a digital camera (QImaging Exi Aqua). The 4X cells were visualized using a 63X water-immersion objective (Olympus, LumiPlan). The cells selected for electrophysiological recordings exhibited a neuron-like morphology with fine-branching neurites. Clusters of amassed cells were avoided. Acquisitions were performed in whole-cell configuration in current-clamp mode using Clampex 10.6 software connected to a Multiclamp 700B amplifier via a Digidata 1550 A digitizer (Molecular Devices). The data were low-pass filtered at 200 Hz and digitized at 10 kHz, and the whole-cell capacitance was compensated. Patch pipettes (resistance of 5–10 MOhm) were filled with an internal solution containing (in mM) 153 K-gluconate, 10 HEPES, 4.5 NaCl, 9 KCl, 0.6 EGTA, 2 MgATP, and 0.3 NaGTP. The pH and osmolarity of the internal solution were close to physiological conditions (pH 7.4, osmolarity of 297 mOsm). The access resistance of the cells in our sample was ~30 MOhm. Among the recordings of 30 neurons that were obtained, 16 were kept for analysis. The rest of the recordings were from neurons that either were not responding to depolarizing steps (putative astrocytes), were unstable, or did not exhibit spontaneous activity; therefore, these recordings were discarded from the analysis.

Spontaneous excitatory postsynaptic potentials (sEPSPs) were recorded in current-clamp gap-free mode (clamped at −45 mV). Current-clamp recordings (at −60 mV) of evoked action potentials were performed by applying a repetitive current pulse (800 ms) with an incremental amplitude (20 pA).

Data analysis was performed using Clampfit 10.6 software (Molecular Devices). To visualize the distribution of the firing frequency, the number of spontaneous spikes per second in current-clamp mode was counted over a one-minute period using the threshold tool of Clampfit software and classified in bins with a width equal to 1 (corresponding to 1 Hz). The data were visualized using the frequency distribution mode of GraphPad Prism V9 software.

### Preparation of organoids slices and electrophysiology

Recordings from organoids were performed between 65 and 80 DIV. The organoids were dipped into a melted aCSF-based low-melting point agarose solution (UltraPure LMP Agarose, InVitrogen, 5%) on a small petri-dish. The aCSF was composed of 125 mM NaCl, 2.5 mM KCl, 2 mM CaCl2, 1 mM MgCl2, 1.25 mM NaH2PO4, 26 mM NaHCO3, 20 mM glucose; at pH 7.4 and oxygenated with 95% O₂ and 5% CO₂. The dish was then transferred on ice for 3 min to solidify the agarose around the organoids. Individual blocks containing the organoids were transferred in the chamber of a vibratom tissue slicer (Leica VT1200), and covered with cold and oxygenated cutting-solution composed of 125 mM NaCl, 2.5 mM KCl, 3 mM MgCl2, 1.25 mM NaH2PO4, 26 mM NaHCO3, 20 mM glucose and 3 mM kynurenic acid, at pH 7.4), at 4 °C.

Sections of 350 μm thickness were prepared at 0.1 mm/s speed and 0.3–0.5 mm amplitude. The agarose slices containing the organoids sections were equilibrated for 30 min at 37 °C in oxygenated aCSF. The organoids were then transferred in the recording chamber at room temperature and were constantly perfused with oxygenated aCSF.

Whole-cell patch-clamp recordings were performed as described in the previous section. The patch-pipettes (resistance 5-7 MOhm) were filled with a biocytin-containing internal solution (in mM): 126 K-gluconate, 10 HEPES, 4 KCl, 4 MgATP, and 0.3 NaGTP, 10 Phospho-creatine, and 8 Biocytin-L-lysine (AnaSpec Inc, Fremont, CA, USA), adjusted to pH 7.4. Neuron-like cells at the periphery of the organoids were selected for recordings. Cells with too few or no spontaneous APs were excluded from the analysis.

### HPLC analysis of dopamine content

At 80 DIV, 1–2 organoids per sample were collected and homogenized in 100 μl of 0.2 M HClO₄. Then, the samples were centrifuged, and the supernatant was collected and spun through a 0.2 μm spin filter (Costar Spin-X, Merck) at 14000 g at 4 °C for 1 min and loaded into an HPLC system (Thermo Scientific Ultimate 3000). The mobile phase was 12.5% acetonitrile buffer (pH 3.0, 86 mM sodium dihydrogen phosphate, 0.01% triethylamine, 2.08 mM 1-octanesulfonic acid sodium salt, and 0.02 mM EDTA). The flow rate of the mobile phase was adjusted to 1.5 ml/min. The dopamine level was calculated using a standard curve generated using external DA standards (the standard curve coefficient of determination was 0.99946). Dopamine content was then normalized to the protein concentration and is expressed in nmol/g.

### Neuronal cultures in microfluidic devices

MSNs and DA neurons were plated in microfluidic devices as previously described[52]. The microfluidic devices were autoclaved and coated with a mixture of poly-D-lysine (0.1 mg/ml) in the MSN and synaptic chambers, and with a mix of poly-D-lysine (0.1 mg/ml) and laminin (10 μg/ml) in the DA chamber overnight at 4 °C. The following day microfluidic devices were washed 3 times with water and then with maturation medium (DMEM-F12/Neurobasal media (2:1) supplemented with N2, retinol-free B27, BDNF, GDNF, DAPT and ascorbic acid) and incubated at 37 °C before neurons were plated. Dissociated MSNs and DA neurons were resuspended in maturation medium, and 150,000 cells were plated in each of the corresponding chambers.

### RV infection

MSNs plated on microfluidic devices were transduced with the tracing vector, a multicistronic lentivirus coexpressing a nuclear GFP, TVA and G-protein necessary for RV infection. Neurons infected with the tracing virus were termed starter MSNs. One week later, the same neurons were transduced with the mCherry expressing EnvA-pseudotyped (G)-deleted RV (dG-RV) and 4 to 5 days postinfection the devices were analyzed by confocal microscope.

### Preparation of hindbrain patterned cells for in vivo transplantation

For each batch of single cells, ten neurospheres from each cell line (H9 and 4X cell lines) were collected and washed twice with PBS at 16 DIV. Then, 500 μl of Accutase (supplemented with 100 μg/ml DNase) was added, and the cells were incubated for 10 min at 37 °C. The neurospheres were first pipetted with a 1 ml pipette followed by a 200 μl pipette to yield a single-cell solution. Five hundred microliters of washing medium (DMEM/F12 supplemented with 1% human serum albumin) was added, and the cells were spun down at 400 g for 5 min at room temperature. The cell pellets were resuspended at a concentration of 100,000 cells/μl in HBSS (supplemented with 100 μg/ml DNase) and kept on ice. The cell suspension was kept on ice for a maximum of three hours, after which a new batch of cells was prepared.

## Preparation of midbrain patterned cells for in vivo transplantation

For single-cell preparations of midbrain patterned progenitors, 16 DIV cultures were used from H9 and 4X cell lines. Cells were detached from the culture well by first washing the cells twice with PBS. Then, 150 µl of Accutase (supplemented with 100 µg/ml DNase) was added, and the cells were incubated for 8–10 min at 37 °C. Following the incubation, cells were pipetted with a 1 ml pipette followed by a 200 µl pipette to yield a single-cell solution. 1 ml of wash medium was used to collect the cells, and the cells were spun down at 400 g for 5 min at room temperature. The cell pellets were resuspended at a concentration of 50,000 cells/µl in HBSS (supplemented with 100 µg/ml DNase) and kept on ice. The cell suspension was kept on ice for a maximum of three hours, after which a new batch of cells was prepared.

## In vivo transplantation of hindbrain patterned progenitors

Adult (9 weeks old) male (225–300 g) ($n = 30$) NIH (NTac:NIH-*Foxn1*[rnu]) nude rats purchased from Taconic Biosciences A/S were group-housed in ventilated cages in a clean room under a 12-h light/dark cycle with *ad libitum* access to sterile food and water. In addition to a standard rat diet, they were given peanuts to increase caloric intake. All animal experiments were conducted in accordance with the guidelines of the European Union Directive (2010/63/EU) and approved by the Danish Animal Inspectorate.

The rats were anesthetized with isoflurane (5% for induction, 2–3% for maintenance), 1.2 L/min of $O_2$, and 0.6 L/min of atmospheric air, and placed in a stereotaxic frame (Stoelting) and unilaterally injected with 6-OHDA (Sigma-Aldrich A/S) (a total of 12 µg free base in 3.5 µl in saline containing 0.02% ascorbic acid)[68] into the right MFB (anteroposterior (AP), −4.4; mediolateral (ML), −1.1; dorsoventral (DV), −7.6; tooth bar, -3.3) using a Hamilton syringe with a glass cannula attached. Following injection, the cannula was left in place for 5 min before being slowly retracted. The incision was sutured, and the animals were injected with buprenorphine (0.03 mg/kg, Temgesic) as an analgesic. Once the animals were fully awake, they were placed back into their cages with wet food and 0.005 mg/ml buprenorphine was given for 2 days in the water.

Lesioning efficiency was assessed 3 weeks postsurgery using the amphetamine-induced rotation test, and animals that exhibited >5 rotations/min were used for further experiments. The selected rats were divided into 3 groups with a similar average number of amphetamine-induced rotations: the 6-OHDA lesion (no transplantation) group ($n = 8$), the H9 cell-transplanted group ($n = 9$), and the 4X cell-transplanted group ($n = 10$) (see Supplementary Fig. 11a, b). Four weeks after lesioning, the animals in the H9 cell-transplanted and 4X cell-transplanted groups were stereotaxically injected into the striatum (AP, +0.5; ML, −3; DV, −4.6/4.8) with 250,000 cells of the respective cell type in a volume of 2.5 µl using a protocol similar to the one described above. All three groups were sacrificed 23 weeks postlesioning (i.e., 19 weeks after transplantation). Two transplanted rats did not complete the study and were euthanized due to health issues: one in the 4X cell-transplanted group (week 8 posttransplantation) due to a broken tail and one in the H9 cell-transplanted group due to hindlimb paralysis (week 17 posttransplantation).

## In vivo transplantation of midbrain patterned progenitors

The midbrain patterned in vivo transplantation experiment was performed in the same manner as the hindbrain experiment, except for the following differences. Adult (6–8 weeks old) male (225–300 g, $n = 32$, Crl:NIH-*Foxn1*[rnu]) nude rats were purchased from Charles River and were group-housed and care as indicated above. As above animals that exhibited >5 rotations/min 3 weeks post-lesion were used for further experiments (Supplementary Fig. 12a, b). The selected rats were divided into 3 groups with a similar average number of amphetamine-induced rotations: the 6-OHDA lesion (no

transplantation) group ($n = 8$), the H9 cell-transplanted group ($n = 10$), and the 4X cell-transplanted group ($n = 10$). 125,000 cells in a volume of 2.5 µl were injected per rat using the stereotaxic coordinates AP, +0.6; ML, −3; DV, −4.7/4.9. All three groups were sacrificed 31 weeks postlesioning (i.e., 27 weeks after transplantation), except one rat from each group which was sacrificed 29 weeks post-transplantation. Two transplanted rats did not complete the study and were euthanized due to health issues: one in the 4X group and another in the H9 group.

## Amphetamine-induced rotation test

The amphetamine-induced rotation test was performed as described previously[54] one week prior to transplantation- to assess the effects of the MFB-lesions - and 8, 18 and 26 weeks posttransplantation. The animals were intraperitoneally (i.p.) injected with 5 mg/kg (hindbrain patterned study) or 3 mg/kg (midbrain patterned study) D-amphetamine and connected to a rotameter (LE 902, PanLab, Harvard Apparatus) coupled to an LE 3806 Multicounter (PanLab, Harvard Apparatus). The number of body rotations over a period of 80 min -starting 10 min after the injection- was recorded. The data are expressed as the net number of full body turns per minute, with ipsilateral rotations having a positive value and contralateral rotations having a negative value. Animals exhibiting >5 turns/min were considered successfully lesioned. One rat had a technical issue during one of the rotation tests and was excluded from this behavioral test.

## Cylinder test

The cylinder test was used to assess paw use asymmetry three weeks postlesioning (one week prior to transplantation) and 18 weeks or 26 weeks posttransplantation for the hindbrain- and midbrain patterned experiments respectively. The animals were placed in a transparent Plexiglas cylinder (height of 30 cm, diameter of 20 cm), and two mirrors were placed behind the cylinder so that the cylinder surface could be fully visualized. Spontaneous activity was video recorded for a total of 5 min. Data analysis was performed by a researcher blinded to the groups using VLC Media Player software in slow motion as previously described[69]. The following behaviors were scored to determine the extent of forelimb-use asymmetry[69]: (a) independent use of the left or right forelimb when touching the wall during a full rear or landing on the floor after a rear (without wall touching) and (b) simultaneous use of both the left and right forelimb to contact the wall of the cylinder during a full rear, for lateral movements along the wall (wall stepping) and for landing on the floor following a rear (without wall touching). All paw touches were analyzed during the first 2 min. If an animal did fewer than 20 paw touches within this time, the full 5 min were analyzed. If a rat had <20 paw touches in 5 min it was considered a failed test, and the rat was excluded from the statistical analysis. This resulted in the exclusion of one rat in the 6-OHDA group in the hindbrain patterned study, and two rats in the 6-OHDA group, three in the H9 group, and two in the 4X group in the midbrain patterned study. The data are presented as the percentage of time each forelimb (left or right) or both forelimbs were used relative to all movements (Left + right + both).

## Immunohistochemical analysis of brain slices

The rats were killed by an overdose of pentobarbital (50 mg/kg i.p.). During respiratory arrest, they were perfused through the ascending aorta with ice-cold saline followed by 4% cold PFA (in 0.1 M NaPB, pH 7.4). The brains were extracted, postfixed in PFA for 3 h, and transferred to 25% sucrose solution (in 0.02 M NaPB) overnight. The brains were cut into 35 µm thick coronal sections with a freezing microtome (Microm HM 450, Brock and Michelsen), separated into serial coronal sections (series of 8 for the striatum and the substantia nigra), and stored at −20 °C.

Immunohistochemical staining was performed on free-floating brain sections using the following primary antibodies: mouse anti-TH

(1:2000, Merck Millipore, cat#MAB318), rabbit anti-TH (1:1000, Pel-Freeze, cat# P40101-150), rabbit anti-GIRK2 (1:500, Alomone, cat#APC-006), mouse IgG1 anti-CALB1 (1:5000, SWANT, cat#300), mouse IgG1 anti-HNA (1:200, Abcam, cat#151181), goat anti-FOXA2 (1:200, R&D Systems, cat#AF2400), sheep anti-hCOL1A1 (1:200, R&D Systems, cat#AF6220), rabbit anti-hCOL3A1 (1:1000, NovusBio, cat# NB120-6580), rabbit anti-EN1 (1:50, Sigma-Aldrich, cat# HPA073141), mouse anti-EN1 (1:2000, gift from Novo Nordisk) rabbit anti-LMX1A (1:5000, Millipore, cat#AB10533), rabbit anti-GABA (1:1000, Sigma-Aldrich, cat#A2052) and mouse anti-hSYP (1:750, Enzo LifeSciences, cat#ADI-905-782-100), 5-HT (1:15,000, Sigma-Aldrich, S5545).

Immunohistochemistry was performed as previously described[68] with avidin-biotin-peroxidase complex (ABC Elite, Vector Laboratories) and 3,3-diaminobenzidine (DAB) as a chromogen to visualize the signal. The sections were mounted on chrome-alum gelatin-coated slides, dehydrated, and coverslipped. The slides were analyzed using an Olympus VS120 Slide Scanner (upright widefield fluorescence) with a 20x objective.

For immunofluorescence, free-floating sections were blocked in 5% normal donkey serum in 0.25% Triton X-100 in KBPS and then incubated overnight with the selected primary antibody in 2.5% donkey serum and 0.25% Triton X-100 in KPBS at room temperature. The sections were washed with KPBS, preblocked for 10 min in 1% donkey serum and 0.25% Triton X-100 in KPBS and incubated for 2 h with corresponding fluorochrome-conjugated secondary antibodies made in donkey (1:200, Jackson ImmunoResearch Laboratories or 1:1000, Invitrogen). DAPI (1:2000, Sigma-Aldrich A/S) was used for nuclear staining. The sections were mounted on chrome-alum gelatin-coated slides with Dako fluorescent mounting medium.

### Microscopic analysis (TH-positive cell number and yield and graft volume)

Coronal sections (1:8) from each animal were immunostained for TH, and DA neurons in the graft were analyzed. Adjacent coronal sections from each animal were used for HNA immunostaining and analysis of surviving cells per transplant. An Olympus VS120 Slide Scanner (upright widefield fluorescence) (Bioimaging Core Facility, Aarhus University) was used to acquire images of the slides using a 20x objective. All sections with visible grafts were selected, and the area in which the number of TH-positive or HNA-positive cells was quantified included the striatum and globus pallidus, but positive cells in the cortex and corpus callosum were not included. The images were analyzed by identifying cells in the region of interest (ROI) using QuPath software[70]. The settings were adapted for each section depending on the staining, and the following settings were used for TH sections: detection image = optical density sum, requested pixel size = 0.5 μm, background radius = 15–30 μm, threshold = 0.15–0.3, median filter radius = 0–3 μm, sigma = 0.7–2 μm, minimum area = 85–130 μm², maximum area = 500–1200 μm², max background intensity = 2, cell expansion = 2 μm. For HNA sections settings were as follows: detection image = optical density sum, requested pixel size = 0.5 μm, background radius = 10 μm, threshold = 0.2, median filter radius = 0.5 μm, sigma = 1.5–2 μm, minimum area = 5 μm², maximum area = 300 μm², max background intensity = 2, cell expansion = 0 μm. The cells were classified by shape, including that of the cell nucleus, and the boundaries were smoothed. With these settings, most cells or nuclei were successfully identified, and these automated counts were also manually inspected and adjusted when necessary. These setting were used across all sections to minimize any bias.

To estimate the number of cells in a full graft, the total number of TH-positive or HNA-positive cells per animal was determined with QuPath software and multiplied by 8, and the Abercrombie method[71] was used to correct for double counting of cells spanning more than one section. The Abercrombie factor of each group was calculated as the average thickness per section divided by (the averaged thickness + the average TH-positive or HNA-positive cell size). These numbers were calculated by sampling 3 sections and 18 cells per animal from 3 different animals per group. The total number of cells in a graft was calculated as the Abercrombie factor x the total number of TH-positive or HNA-positive cells x8. The number of surviving cells that became TH-positive cells (yield of TH-positive cells) was estimated per 100,000 transplanted cells. The volume of each TH graft was estimated using the Cavalieri's principle with $V = A1T1 + A2T1 + \ldots + A_nT1$, where V is the estimated volume, T1 is the sampling interval of a 1/8 series (i.e., $8 \times 35\,\mu m$), and A(n) is the TH-positive area in the respective section (n)[72]. The percentage of HNA cells that were also TH-positive cells was calculated by dividing the estimated number of TH-positive cells by the estimated number of HNA-positive cells and multiplied by 100 (total number of TH-positive cells/total number of HNA-positive cells x 100).

For 5-HT analyses, coronal sections (1:8) from five animals from each transplanted group were immunostained and the number of serotonergic neurons were manually counted using a 10x objective. The Abercrombie method was used to correct for double counting as previously described, and the percentage of HNA cells that were also 5-HT positive was calculated as described above with TH.

### Fiber outgrowth analyses

Using an Olympus VS120 Slide Scanner (Bioimaging Core Facility, Aarhus University) and OlyVIA software, images of the TH-stained coronal sections were taken at x15 magnification in the striatal area outside the body of the graft, where no cell bodies were present. Two sections with graft per animal were used (separated by 560 μm and located between the AP coordinates −0.26 and 1.70 mm from bregma) and three images were captured for each striatal section. ImageJ software (U.S. National Institutes of Health) was used to convert the images to 32-bit greyscale images and to measure the area fraction of the images covered by TH positive immunostaining. The average of the area fractions of the two grafted sections was then calculated for each animal and later average per group.

### Statistical analysis

All statistical analyses were performed using GraphPad Prism v 9. One-way ANOVA or two-way ANOVA was performed, and Tukey or Sidak's test were used for post hoc analysis when appropriate. Unpaired, two-tailed $t$ tests were used when comparing only the grafted groups. All data are presented as the mean ± standard error of the mean (SEM) or ±standard deviation (SD) (as indicated). $P < 0.05$ was considered significant.

### Reporting summary

Further information on research design is available in the Nature Portfolio Reporting Summary linked to this article.

## Data availability

The RNA-seq data generated from this study have been deposited in NCBI's Gene Expression Omnibus with the accession number GEO: GSE227071. Source data are provided with this paper. The Human genome build hg38 (Ensembl release 92) is available at [https://www.ensembl.org/Homo_sapiens/Info/Index]. Source data are provided with this paper.

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

## Acknowledgements

We thank Susanne Hvolbøl Buchholdt Seldrup for assistance with culturing and maintaining the pluripotent stem cell lines and Sanne Nordestgaard Andersen for assistance with differentiation experiments. We thank Gitte Ulbjerg Toft for excellent technical help with the in vivo rat experiments and postmortem histological analysis of the brains. We would also like to thank Per Fuglsang Mikkelsen for assistance with HPLC and Ilaria Rossomanno for her help in the in vitro quantitative analysis. We thank Wen-Hsien Hou for technical help with electrophysiology recordings. We thank Veronica Castiglioni for technical help with flow cytometry. The technical expertize of Jane Pedersen regarding karyotyping is highly appreciated. We are grateful for the assistance and use of the AU Health Bioimaging Core Facility, Animal Facility, Bioinformatics Core Facility and FACS Core Facility. Funding: This study was supported by Lundbeckfonden (grant no. DANDRITE-R248-2016-2518 (M.D., A.N.) and R248-2017–431 (A.N.)), the Parkinsonforeningen (M.D.), and The Danish National Research Foundation grant no. DNRF133 (A.N.). M.D. is a partner of BrainStem—Stem Cell Center of Excellence in Neurology, funded by Innovation Fund Denmark.

## Author contributions

Conceptualization: M.D.; methodology: M.M., M.C., R.K., N.M.J., D.N., J.R.C., J.C.N., M.R.-R., M.D.; formal analysis: M.M., M.C., F.F., N.M.J., D.N., J.L., E.U., I.H.K., J.R.C., J.C.N., R.K., A.I., U.B.J., P.Q., M.R.-R., M.D.; investigation: M.M., M.C., F.F., R.K., N.M.J., D.N., J.L., E.U., I.H.K., J.R.C., J.C.N., A.I., U.B.J., P.Q., M.R.-R., M.D.; resources: S.N., V.B., M.R.-R., M.D.; writing - original draft: M.R.-R., M.D.; writing - review & editing: M.C., F.F., R.K., N.M.J., D.N., J.L., J.C.N., E.U., P.Q., V.B., A.N., M.R.-R., M.D.; visualization: M.M., M.C., R.K., N.M.J., D.N., J.L., E.U., J.R.C., J.C.N., A.I., M.R.-R., M.D.; supervision: A.N., S.N., V.B., M.R.-R., M.D.; funding acquisition: A.N., M.R.-R., M.D. All authors read and approved the final manuscript.

## Competing interests

M.M. and M.D. are inventors on a patent application filed by Aarhus University, WO2022136306A1 (National phase) that is related to the generation of lineage restricted cells. J.C.N. and J.R.C. are employed by Novo Nordisk. The remaining authors declare no competing interests.
