## [Peer Review File · Nature Communications]

REVIEWER COMMENTS

Reviewer #1 (Remarks to the Author):

Maimaitili et al. demonstrated that lineage restriction could control cell fate and generate functional mesencephalic dopaminergic (DA) neurons. They developed a human embryonic stem cell (ESC) line called 4X LR-USCs (lineage restricted-undifferentiated stem cells) by knocking out genes that are involved in the early specification of the hindbrain (GBX2) and spinal cord (CDX1/2/4). They showed that the 4X LR-USCs could differentiate efficiently to caudal midbrain floor plate progenitors, which is positive for FOXA2/LMX1A/OTX2/EN1 in midbrain inducing culture condition but also in the hindbrain-inducing culture condition. They also showed that the LR-USCs-derived DA neurons were electrophysiologically active and restored the motor function of Parkinson's disease model rats. The authors claimed that the strategy of lineage restriction by gene editing approach could prevent the induction of unwanted lineages and enhance the generation of target cells, which can be useful for regenerative medicine such as cell transplantation or drug screening. The approach shown in the manuscript can be one of the strategies for next-generation stem cell therapy for Parkinson's disease. Still, I have to say there is no advantage or novelty of using the LR-USCs for the reasons below.

- 1) Importantly, the result of the hindbrain and spinal cord gene knockout is predictable and conceptually not novel.
- 2) Using a hindbrain-inducing protocol, the authors showed the advantage of 4X LR-USCs in vitro and in vivo. As an appropriate control, however, it is required to show the benefit of the new strategy over the current method using a midbrain-inducing protocol.
- 3) The authors claimed 4X-LR-USCs could prevent unwanted cells in the graft. However, the unwanted cells in the current protocol for mesencephalic dopaminergic neurons are choroid plexus cells or perivascular fibroblasts, not hindbrain or spinal cord cells (Doi et al. Nat Commun 2020, Tiklova et al. Nat Commun 2020).

Reviewer #2 (Remarks to the Author):

In this study by Maimaitili et al, the authors describe the generation of lineage-restricted human undifferentiated stem cells (LR-USCs) and their subsequent differentiation using midbrain and hindbrain conditions.

The characterization of mesencephalic dopaminergic (mesDA) neurons from LR-USCs is extensive using a range of cellular, molecular, and functional assays, and the results are clearly presented. Importantly, the authors validate their findings across multiple hPSC lines.

Altogether, the manuscript is well-written, the experiments are well-designed, and the figures are clear. Although the manuscript contains an impressive amount of work with the thorough characterization of differentiation experiments, there is inconsistency in the differentiation paradigms used (multiple different hindbrain conditions at various time points) which makes the interpretation of the data and comparison difficult. This includes both the use of different concentrations of Chir as also the variation between 2D and 3D models. Another major issue of the manuscript is that it is unclear if the LR-USCs would in fact provide any clinical benefit upon transplantation if differentiated using midbrain conditions (rather than hindbrain conditions). Indeed, no cellular therapy would be performed using cells that have undergone hindbrain differentiation in vitro. To address this, the authors could perform their in vivo transplantation studies comparing LR-USCs to WT hPSCs following midbrain differentiation paradigms.

There are some changes that I believe would strengthen the manuscript:

Major critiques:

1. Fig 1f: the authors conclude that 4x cells were unable to generate progenitor cell types caudal to r4. Based on the available RNA-seq data and brief differentiation experiment, this conclusion may be too strong. Differentiation to more caudal progenitor cells could also simply be delayed. It would be helpful to perform a similar experiment at a later time point to exclude this possibility.
2. Line 192-194 and Fig 1f: the authors state that “In contrast, ... loss of posterior HOX expression.” This is not fully clear, as the 4x cells are also KO for GBX. Why these cells then show a similar, or perhaps even increased differentiation to anterior hindbrain identity (compared to WT) is confusing. The authors should provide a better explanation for this.
3. Line 196-197: “cells include HOX genes (Fig. 1g).” The authors should be more specific here, and state that this relates to posterior HOX genes.
4. Following the data of Fig 1 and the conclusion in line 231-233 that 4x cells are lineage-restricted and preferentially adopt a midbrain or anterior hindbrain identity, the authors should provide more explanation on why 4x cells preferentially differentiate into anterior hindbrain compared to GBX KO cells. This is a curious phenomenon which is not easily understood based on the available data and developmental literature.
5. Fig 2e and f: The authors describe the development of a flow cytometry panel to simultaneously examine the protein expression of FOXA2, OTX2, LMX1A, and EN1, also used in Fig 2c. The authors should use this same panel for experiments performed for 2e and f.

6. There is no clear rationale why, for the data of Fig 3, the authors used an alternative hindbrain differentiation strategy compared to previous figures. Initially, the authors used Chir concentrations of 3uM, whereas now they use Chir concentrations of 1uM. This makes comparison of the data difficult.
7. Fig 3g: considering the interesting discrepancy between anterior and posterior HOX gene expression (in previous hindbrain diff experiments, fig 1) in the 4x cells, it would be of value if the authors could provide a more detailed analysis on this for the H9 and 4x cells.
8. It is unclear when the authors switch from a 2D to a 3D differentiation paradigm (considering the description of spheres in the legend of Fig 3 and the ICC data of Fig S7). This is somewhat confusing and should be better described.
9. Based on the initial findings from Fig 1, it is unsurprising that the scRNA-seq data in Fig 3 and 4 show an increased mesDA neuron differentiation efficiency of the 4x cells compared to the WT cells under hindbrain conditions. However, to understand their utility, it would perhaps have been more informative to understand the differences between 4x and WT cells when differentiated under midbrain conditions.
10. In line 360 there is already reference to the data of fig S74, which are clearly derived from 3D organoids. However, later in the manuscript, in line 398, it states "...but adapted it to generate organoids to provide an optimal environment for the survival of neurons." The order of the data and introduction of this 3D model needs to be better structured.
11. For data of figure 5H-J the authors switch back to midbrain condition differentiations.
12. Line 236-237: "Our main goal... mesDA neurons than hPSCs." Considering this is the authors' main goal, it is unclear why the authors decide to use a hindbrain differentiation paradigm for the in vivo transplantation experiments. To test their downstream use and value, it would be much more important to compare LR-USCs to WT hPSCs when differentiated under midbrain conditions. While the authors provide a possible explanation in line 458 "rather than perform a benchmarking experiment", they propose the opposite for earlier experiments, see line 237: "Thus, we performed a benchmarking study and compared...". As a whole, the in vivo experiments are technically well characterized but lack value due to the differentiation conditions tested, as midbrain conditions are not included.
13. In summary, for the data of Fig 3, 4, and 6 it would greatly benefit the study to include comparisons between the 4x and WT hPSC cells differentiated under midbrain conditions. For a claim as in line 622: "This has significant advantages for clinical applications," such experiments would be necessary.

Minor critiques:

1. Fig S1: the authors should provide further characterization of hPSCs after gene-editing on pluripotency, differentiation into all three lineages, potential karyotypic abnormalities and off-target effects. Why did the authors not also do CNV analysis here, as in Fig S2?
2. Fig 1C: how was quantification done? How many independent experiments? Please show results of quantification.

3. Fig 1i: inconsistent choice of data representation, between RNA count and mRNA fold change. Please provide rationale for choices or make consistent between all data.
4. Fig S2: the authors should provide further characterization of hPSCs after gene-editing on pluripotency and differentiation into all three lineages. Did the authors check for CDX1 and CDX4 transcripts in 4x cell-derived CNPs?
5. Fig 2f: please keep the y-axis consistent between experiments, as this can be misleading (see data for EN1/OTX2 cells for GBA lines).
6. Line 208-211: "We found that the... respectively; Fig. 1i)." These data are not significantly different between the conditions, and the authors would be advised to place less emphasis on this in the text.
7. Fig 1h and i, Fig S5: why did the authors decide to switch to H1 line for the main figure? Either show data for both lines in the main figure or be consistent with previous figures.
8. The addition of ICC data at DIV16 as validation of the scRNA-seq analyses to the main figure 3 (from S6-7) would be beneficial.
9. Fig 5j: would it be possible to quantify the connectivity (mCherry+ neurons) for both genetic backgrounds?

Reviewer #3 (Remarks to the Author):

In this paper, the authors generated human ES cells in which four genes were knocked out using the CRISPR/Cas9 method, and analyzed in detail using the scRNA-seq method that these cells lack the ability to give rise to hindbrain and spinal cord progenitors and efficiently induce midbrain progenitors. Furthermore, by transplanting these cells into parkinsonian rat model, the authors confirmed behavioral improvement and an increase in the number of survival dopaminergic neurons.

This study was well designed based on the original concept of increasing the efficiency of induction in the midbrain region by deficient hindbrain induction ability, and the results are expected to improve the efficacy of cell transplantation therapy, which is beneficial for improving the viability of transplanted cells. The quality of the data is very high, and the data is accurate and solidly collected.

Here is my comment.

The authors showed that mesencephalic cells appear efficiently after transplantation of cells induced from 4x cells, but I think it is necessary to confirm if hindbrain neurons such as serotonergic neurons are contaminated in the graft. Since the manuscript only shows data from in vitro study, I think verification is needed regarding the ability to induce hindbrain neurons such as serotonergic neurons in vivo.

REVIEWER COMMENTS

Reviewer #1 (Remarks to the Author):

Maimaitili et al. demonstrated that lineage restriction could control cell fate and generate functional mesencephalic dopaminergic (DA) neurons. They developed a human embryonic stem cell (ESC) line called 4X LR-USCs (lineage restricted-undifferentiated stem cells) by knocking out genes that are involved in the early specification of the hindbrain (GBX2) and spinal cord (CDX1/2/4). They showed that the 4X LR-USCs could differentiate efficiently to caudal midbrain floor plate progenitors, which is positive for FOXA2/LMX1A/OTX2/EN1 in midbrain inducing culture condition but also in the hindbrain-inducing culture condition. They also showed that the LR-USCs-derived DA neurons were electrophysiologically active and restored the motor function of Parkinson's disease model rats. The authors claimed that the strategy of lineage restriction by gene editing approach could prevent the induction of unwanted lineages and enhance the generation of target cells, which can be useful for regenerative medicine such as cell transplantation or drug screening. The approach shown in the manuscript can be one of the strategies for next-generation stem cell therapy for Parkinson's disease. Still, I have to say there is no advantage or novelty of using the LR-USCs for the reasons below.

Comment:

1) Importantly, the result of the hindbrain and spinal cord gene knockout is predictable and conceptually not novel.

Response:

I appreciate the comments; however, I would like to mention the novelty of our work has been confirmed by a patent office. Furthermore, reviewer 3 states, *"This study was well designed based on the original concept of increasing the efficiency of induction in the midbrain region by deficient hindbrain induction ability,"*

Comment:

2) Using a hindbrain-inducing protocol, the authors showed the advantage of 4X LR-USCs in vitro and in vivo. As an appropriate control, however, it is required to show the benefit of the new strategy over the current method using a midbrain-inducing protocol.

Response:

We agree with the review, and indeed, we have shown in vitro data under midbrain conditions (see figure 2). In vitro we show that when LR-USCs were differentiated under

midbrain conditions, 68.7% of the cells were caudal midbrain floor plate progenitors, i.e., were quadruple-positive for FOXA2/LMX1A/OTX2/EN1, whereas only 24.8% of the cells generated from human embryonic stem cells (hESCs) were caudal midbrain floor plate progenitors (Fig. 2).

Additionally, we have now included a transplantation experiment performed with hESCs and LR-USCs differentiated with a midbrain protocol (Figure 7). We show a higher yield of DA neurons and faster recovery with the 4X cells. Importantly, we transplanted only 125,000 cells per rat, and we showed that only the rats containing 4X cells showed improvements in the cylinder test. The cylinder test is a drug-free test that assesses if the transplanted dopamine neurons are functionally integrated. This is a clinically relevant behavioral test that directly addresses the concern regarding clinical benefit. In sum, we show LR-USCs generate a higher purity that results in a graft with higher potency, resulting in a functional behavioral benefit.

Comment:

3) The authors claimed 4X-LR-USCs could prevent unwanted cells in the graft. However, the unwanted cells in the current protocol for mesencephalic dopaminergic neurons are choroid plexus cells or perivascular fibroblasts, not hindbrain or spinal cord cells (Doi et al. Nat Commun 2020, Tiklova et al. Nat Commun 2020).

Response:

Firstly, the aim is not exclusively to prevent hindbrain populations, but more importantly, by knocking out this region, we expand the midbrain, increasing the efficiency of generating dopaminergic neurons. We show that with our 4X line, they are more robust at generating midbrain cells when differentiated in midbrain and hindbrain conditions (Fig. 2, Fig 3, Fig. 7). Secondly, current midbrain protocols show that after grafting, only a minor component of the graft is TH positive. On average, only 10% of the grafted cells become dopaminergic neurons. For example, from Parmar's lab, the FGF8 protocol (Kirkeby et al., CSC, 2017; Nolbrant et al., Nature Protocols 2017) states that 3,700 cells are TH positive per 100,000 grafted and the TH/HNA% = 11.8%. Kim et al., CSC (2021) obtained 2,038 DA neurons per 100,000 grafted cells. Nolbrant et al., Nature Protocols (2017) generated 3,700 TH-positive cells per 100,000 grafted cells. From Takahashi's group, they generated 1,150 TH-positive cells per 100,000 (Hiramatsu et al., JPD 2022). Overall, most protocols in the field generate a low yield of dopaminergic neurons post-grafting. Our LR-USCs generate significantly more DA neurons in our in vivo midbrain experiment (7,488 TH-positive cells per 100,000 cell grafted vs. hESCs-derived DA neurons 4,250; Fig 7f).

Thirdly, the unwanted cells are not limited to choroid plexus or perivascular fibroblasts. Fiorenzano et al. Nat Commun 2021, showed that by single-cell sequencing, a large proportion of the neurons are non-dopaminergic and match those from the oculomotor and trochlear nucleus (see Fiorenzano Supplementary fig. 5d; below).

Furthermore, in Kirkeby et al., CSC, 2017 paper, they show their progenitor populations express not only midbrain markers but also high levels of *FGF8*. *FGF8* is expressed in the hindbrain region of the isthmus organiser, confirming that indeed hindbrain contaminating progenitor cells are generated from this protocol (see image below from Kirkeby et al., CSC, 2017 Fig. 1d,e).

Additionally, the publication by Xu et al., JCI (<https://doi.org/10.1172/JCI156768>) used single-cell analysis and identified several populations posterior to the midbrain, including Isthmic organiser cells, and hindbrain populations. In their paper, they write: “We found that this process recapitulated the development of multiple but adjacent fetal brain regions including the ventral midbrain, the isthmus, and the ventral hindbrain, resulting in a heterogenous donor cell population.” See images from Fig 3 below.

Reviewer #2 (Remarks to the Author):

In this study by Maimaitili et al, the authors describe the generation of lineage-restricted human undifferentiated stem cells (LR-USCs) and their subsequent differentiation using midbrain and hindbrain conditions.

The characterization of mesencephalic dopaminergic (mesDA) neurons from LR-USCs is extensive using a range of cellular, molecular, and functional assays, and the results are clearly presented. Importantly, the authors validate their findings across multiple hPSC lines.

Altogether, the manuscript is well-written, the experiments are well-designed, and the figures are clear. Although the manuscript contains an impressive amount of work with the thorough characterization of differentiation experiments, there is inconsistency in the differentiation paradigms used (multiple different hindbrain conditions at various time points) which makes the interpretation of the data and comparison difficult. This includes both the use of different concentrations of Chir as also the variation between 2D and 3D models. Another major issue of the manuscript is that it is unclear if the LR-USCs would in fact provide any clinical benefit upon transplantation if differentiated using midbrain conditions (rather than hindbrain conditions). Indeed, no cellular therapy would be performed using cells that have undergone hindbrain differentiation in vitro. To address this, the authors could perform their in vivo transplantation studies comparing LR-USCs to WT hPSCs following midbrain differentiation paradigms.

There are some changes that I believe would strengthen the manuscript:

Major critiques:

Comment:

1. Fig 1f: the authors conclude that 4x cells were unable to generate progenitor cell types caudal to r4. Based on the available RNA-seq data and brief differentiation experiment, this conclusion may be too strong. Differentiation to more caudal progenitor cells could also simply be delayed. It would be helpful to perform a similar experiment at a later time point to exclude this possibility.

Response

We have now extended the differentiation to 32 days (supplementary Fig. 3). With the extended differentiation, we confirmed the same results as we did at 11 days. Specifically, at 32 days of differentiation, we also see the absence of cells expressing HOX markers caudal to r4.

Comment:

2. Line 192-194 and Fig 1f: the authors state that “In contrast, ... loss of posterior HOX expression.” This is not fully clear, as the 4x cells are also KO for GBX. Why these cells then show a similar, or perhaps even increased differentiation to anterior hindbrain identity (compared to WT) is confusing. The authors should provide a better explanation for this.

Response:

The reason there is an increase in *MAFB* expression in 4X CNPs is due to several reasons. Firstly, the CNP protocol generates a cell population that contains cells spread across a

broad region of the hindbrain and spinal cord. In the GBX2^{-/-} line, the GBX2 knockout affects R1-3, and the cells can still be spread across a large region of the hindbrain and spinal cord. Whereas in the 4X line, the additional CDX1/2/4 KO removes R5 onwards. Therefore, there is only a small region of the remaining hindbrain (R4) remaining. Secondly, CDX1 is a known repressor of MAFB, and loss of CDX1 can cause ectopic expression of MAFB in this remaining hindbrain region.

To make this more clear in the manuscript, we have now modified the sentence to include the reference regarding CDX1: Line 193: “..., which is in accordance with the loss of CDX1 and posterior HOX expression (Sturgeon 2011)”. Furthermore, on Line: 235 we state: “...4X cells did not produce spinal cord progenitors and showed a restricted HOX expression profile up to r4.”

Comment:

3. Line 196-197: “cells include HOX genes (Fig. 1g).” The authors should be more specific here, and state that this relates to posterior HOX genes.

Response

We have now modified the sentence and stated posterior HOX genes. Line 197: “...included posterior HOX genes (Fig. 1g).”

Comment:

4. Following the data of Fig 1 and the conclusion in line 231-233 that 4x cells are lineage-restricted and preferentially adopt a midbrain or anterior hindbrain identity, the authors should provide more explanation on why 4x cells preferentially differentiate into anterior hindbrain compared to GBX KO cells. This is a curious phenomenon which is not easily understood based on the available data and developmental literature.

Response:

We have now explained this clearly within that section of the manuscript. Specifically, on lines 205-208, we added: *Consequently, when 4X is differentiated using a protocol that produces posterior hindbrain and spinal cord cells, it cannot generate these cell types. Instead, 4X cells adopt an alternate fate of a more anterior identity, specifically midbrain cells or the remaining region of hindbrain not regulated by the four genes.*

Comment:

5. Fig 2e and f: The authors describe the development of a flow cytometry panel to simultaneously examine the protein expression of FOXA2, OTX2, LMX1A, and EN1, also used in Fig 2c. The authors should use this same panel for experiments performed for 2e and f.

Response:

We have now completed the flow cytometry analysis on our differentiation experiments performed in 2e and f and updated those graphs. As expected, the results show that all 4X lines produced significantly higher percentage of quadruple-positive cells. See Lines 277-285.

Comment:

6. There is no clear rationale why, for the data of Fig 3, the authors used an alternative hindbrain differentiation strategy compared to previous figures. Initially, the authors used Chir concentrations of 3uM, whereas now they use Chir concentrations of 1uM. This makes comparison of the data difficult.

Response:

We changed between 3uM to 1uM because the 3uM concentration generates posterior hindbrain and spinal cord cells (as caudal as HOXA9 expressing cells). This protocol was aimed to test the posterior limits of 4X because of the KO of CDX genes. However, for experiments in Fig. 3 we were interested in patterning the cells to an anterior hindbrain region closer to the midbrain. The anterior hindbrain is regulated by GBX2, and this protocol tests the lineage restriction in this context. Furthermore, when 4X cells are differentiated using an anterior hindbrain protocol, we show that they still generate midbrain cells. This is a rigorous test that shows the effectiveness of lineage restriction. 4X cells have a clear advantage because they robustly generate midbrain cell types under a broader range of conditions.

To make this clear in the text, we have now written in lines 311-314: *“In the first experiments we used 3μM which in hESCs primarily generate posterior hindbrain and spinal cord cells that are regulated by CDX genes (Fig. 1f). In this experiment we patterned the cells to an anterior hindbrain region which is controlled by GBX2. Thus, we used a GSK3i concentration of 1μM (Fig 3a).”*

Comment:

7. Fig 3g: considering the interesting discrepancy between anterior and posterior HOX gene expression (in previous hindbrain diff experiments, fig 1) in the 4x cells, it would be of value if the authors could provide a more detailed analysis on this for the H9 and 4x cells.

Response:

We have now included the expression of each *HOXA* and *HOXB* genes in each cluster separated for H9 and 4X cells (see supplementary figure 8).

Comment:

8. It is unclear when the authors switch from a 2D to a 3D differentiation paradigm (considering the description of spheres in the legend of Fig 3 and the ICC data of Fig S7). This is somewhat confusing and should be better described.

Response:

We have now added to the schematic diagrams a section showing when the cells are grown as 2D or 3D (see Fig 1h, Fig 2a,g, and Fig 3a)

Comment:

9. Based on the initial findings from Fig 1, it is unsurprising that the scRNA-seq data in Fig 3 and 4 show an increased mesDA neuron differentiation efficiency of the 4x cells compared to the WT cells under hindbrain conditions. However, to understand their utility, it would perhaps have been more informative to understand the differences between 4x and WT cells when differentiated under midbrain conditions.

Response

We agree with the reviewer, and indeed, all the data in figure 2 are results from midbrain differentiation conditions. In addition, we have now performed an *in vivo* transplantation experiment performed using midbrain conditions for H9 and 4X (see Fig. 7).

We show higher yield of DA neurons and faster recovery with the 4X cells. Importantly, we transplanted only 125,000 cells per rat, and we showed that only the rats containing 4X cells showed improvements in the cylinder test. The cylinder test is a drug-free test that assesses if the transplanted dopamine neurons are functionally integrated. This is a clinically relevant

behavioral test that directly addresses the concern regarding clinical benefit. In sum, we show LR-USCs generate a higher purity that results in a graft with higher potency, resulting in a functional behavioral benefit.

Comment:

10. In line 360 there is already reference to the data of fig S74, which are clearly derived from 3D organoids. However, later in the manuscript, in line 398, it states "...but adapted it to generate organoids to provide an optimal environment for the survival of neurons." The order of the data and introduction of this 3D model needs to be better structured.

Response:

The data shown in supplementary fig. 7f (now moved to Fig3), are cultured in suspension as neurospheres up to day 16. When we extended the differentiation to generate neurons and maintained them in suspension, we referred to them as organoids because they contained mature neurons. As mentioned above, we have now clearly indicated how the cells were grown in the schematic diagrams. It is also described in the methods.

Comment:

11. For data of figure 5H-J the authors switch back to midbrain condition differentiations.

Response:

We were interested in examining the functional aspects of DA neurons derived from 4X cells and comparing them to unedited hESCs. Therefore, we needed to perform the experiment under midbrain conditions so that the unedited hESCs could generate DA neurons.

Comment:

12. Line 236-237: "Our main goal... mesDA neurons than hPSCs." Considering this is the authors' main goal, it is unclear why the authors decide to use a hindbrain differentiation paradigm for the in vivo transplantation experiments. To test their downstream use and value, it would be much more important to compare LR-USCs to WT hPSCs when differentiated under midbrain conditions. While the authors provide a possible explanation in line 458 "rather than perform a benchmarking experiment", they propose the opposite for earlier experiments, see line 237: "Thus, we performed a benchmarking study and compared...". As a whole, the in vivo experiments are technically well characterized but lack value due to the differentiation conditions tested, as midbrain conditions are not included.

Response:

The main value of the experiment is that it clearly demonstrates that under anterior hindbrain conditions, 4X can still robustly generate dopaminergic neurons.

We agree with the reviewer that midbrain experiments are valuable to do. We have now included in vivo transplantation results performed using midbrain differentiated cells from H9 and 4X (see Fig. 7) and as discussed above.

Comment:

13. In summary, for the data of Fig 3, 4, and 6 it would greatly benefit the study to include comparisons between the 4x and WT hPSC cells differentiated under midbrain conditions. For a claim as in line 622: "This has significant advantages for clinical applications," such experiments would be necessary.

Response:

Figure 2 contains in vitro data of the cells differentiated under midbrain conditions. We have now included in vivo differentiation conditions under midbrain conditions (Fig.7).

Minor critiques:

1. Fig S1: the authors should provide further characterization of hPSCs after gene-editing on pluripotency, differentiation into all three lineages, potential karyotypic abnormalities and off-target effects. Why did the authors not also do CNV analysis here, as in Fig S2?

Response:

- We now show the expression of pluripotent genes in the cell lines GBX2^{-/-} and 4X when maintained in the undifferentiated state (see Supp. Fig. 2l)
- We have now differentiated GBX2^{-/-} and 4X line into mesoderm and endoderm (see Supp. Fig. 1l and Supp Fig. 2m).
- We have now performed karyotype analysis on both GBX2^{-/-} and 4X and a trained clinician has confirmed that the karyotypes are normal (see Supp. Fig. 1k and Supp. Fig. 2k).
- We have now included off-target analysis for GBX2^{-/-} (see Supplementary Table 1) and supplementary table 1 also includes the off-target analysis for 4X. No indels were identified at the off-target sites for GBX2^{-/-} or 4X.
- We have now completed the CNV analysis for GBX2^{-/-} and we found no significant difference to the unedited H9 line (see supp Fig 2j).

2. Fig 1C: how was quantification done? How many independent experiments? Please show results of quantification.

Response:

We quantified the percentage of OTX2 and CDX2 positive cells in GBX2^{-/-} day 4 progenitors using FACS. The experiment was repeated three times. The FACS results are shown in Supplementary Fig. 1i,j. and the number of replicates is stated in the supplementary figure. In addition, we used QPCR to quantify the express levels of *OTX2* and *CDX2* in H9, 4X, and CDX2^{-/-} cells at 4 DIV (Fig. 1d). These experiments were repeated in triplicate and stated in the figure legend.

3. Fig 1i: inconsistent choice of data representation, between RNA count and mRNA fold change. Please provide rationale for choices or make consistent between all data.

Response:

The differences here represent different methods for analysing RNA, which include Nanostring and QPCR. We have now added to the figure legend: "Nanostring data shown as RNA count and QPCR as fold change".

4. Fig S2: the authors should provide further characterization of hPSCs after gene-editing on pluripotency and differentiation into all three lineages. Did the authors check for CDX1 and CDX4 transcripts in 4x cell-derived CNPs?

Response:

We have now included in Fig S2 the expression of *CDX1*, *CDX2*, and *CDX4* in 4X and H9-derived CNPs.

5. Fig 2f: please keep the y-axis consistent between experiments, as this can be misleading (see data for EN1/OTX2 cells for GBA lines).

Response:

We have now made the y-axis consistent between experiments (Fig. 2e)

6. Line 208-211: “We found that the... respectively; Fig. 1i).” These data are not significantly different between the conditions, and the authors would be advised to place less emphasis on this in the text.

Response:

We agree with the reviewer and have removed the description of OTX2 expression at day 11. Since completing the longer differentiation to day32 we see that OTX2 is significant at day 32. Therefore, we modified the sentence to refer to day 32.

7. Fig 1h and i, Fig S5: why did the authors decide to switch to H1 line for the main figure? Either show data for both lines in the main figure or be consistent with previous figures.

Response:

We have now included data for both cell lines in the main Fig. 2h.

8. The addition of ICC data at DIV16 as validation of the scRNA-seq analyses to the main figure 3 (from S6-7) would be beneficial.

Response:

We agree with the reviewer, and we have now moved the ICC data to the main figure 3h.

9. Fig 5j: would it be possible to quantify the connectivity (mCherry+ neurons) for both genetic backgrounds?

Response:

This experiment was a qualitative assessment of the neuron’s function. Specifically, we aimed to show that 4X-derived DA neurons could make synaptic connections in vitro. It is technically challenging to control the variability on the infection rates of the multiple viruses to the degree required for accurate quantification. Instead, to assess the connectivity and have a quantitative assessment of their function, we performed in vivo transplantation experiments (Fig 6 and 7). In the in vivo experiments, we used two behavioural experiments (the rotation test and the cylinder test) to examine and quantify the functional aspects of the neurons.

Reviewer #3 (Remarks to the Author):

In this paper, the authors generated human ES cells in which four genes were knocked out using the CRISPR/Cas9 method, and analyzed in detail using the scRNA-seq method that these cells lack the ability to give rise to hindbrain and spinal cord progenitors and efficiently induce midbrain progenitors. Furthermore, by transplanting these cells into parkinsonian rat model, the authors confirmed behavioral improvement and an increase in the number of survival dopaminergic neurons.

This study was well designed based on the original concept of increasing the efficiency of induction in the midbrain region by deficient hindbrain induction ability, and the results are expected to improve the efficacy of cell transplantation therapy, which is beneficial for improving the viability of transplanted cells. The quality of the data is very high, and the data is accurate and solidly collected.

Here is my comment.

The authors showed that mesencephalic cells appear efficiently after transplantation of cells induced from 4x cells, but I think it is necessary to confirm if hindbrain neurons such as serotonergic neurons are contaminated in the graft.

Comment:

Since the manuscript only shows data from in vitro study, I think verification is needed regarding the ability to induce hindbrain neurons such as serotonergic neurons in vivo.

Response:

We agree with the reviewer that investigating serotonergic neurons is important, and we have now done that.

For the first transplantation experiment using hindbrain patterned cells, we identified a few serotonergic neurons. Specifically, we state on lines 558-562: *“Additionally, we counted few 5-HT serotonergic cells across both H9 and 4x grafts. Serotonergic neurons made up 2.07% (± 0.31 SEM) of the HNA-positive cells in the H9 group, and in the 4X group serotonin neurons only made up 0.54% (± 0.14 SEM) of the HNA cells ($P < 0.01$; Supplementary Fig. 11l,m).”*

For the new second transplantation experiment using midbrain patterned cells, we state on lines 598-599: *“In both H9 and 4X cell grafts, we rarely detected serotonergic neurons (between 1-11 5HT-positive cells detected across a series of 8).”*

REVIEWERS' COMMENTS

Reviewer #1 (Remarks to the Author):

The authors added transplantation experiments with cells differentiated by a midbrain-inducing protocol in the revised manuscript. The results showed a higher yield of DA neurons and faster recovery with the 4X-cells, and only rats with 4X-cells showed improvements in the cylinder test when only 125,000 cells were transplanted.

In the revised Figure 6, the authors showed the advantage of 4X-graft in the brain of 6-OHDA lesioned rats. There are more TH-positive cells in the 4X-cell graft, but the ratio of TH in HNA is not different from the control group. These results suggest that the 4X-cells survived and/or proliferated more than the control cell line. Cell survival in the brain is one of the critical issues to be solved in regenerative therapy, so the results benefit this research field. I recommend adding some interpretations of the mechanism of the better survival of 4X-cells in the brain in the discussion.

Overall, the revised manuscript shows more clearly the advantage of 4X-cells. I am almost satisfied with the answers by the authors, except for one point mentioned above.

Reviewer #2 (Remarks to the Author):

In this study by Maimaitili et al, the authors describe the generation of lineage-restricted human undifferentiated stem cells (LR-USCs) and their subsequent differentiation using midbrain and hindbrain conditions. This approach improves yields of DA neurons under both midbrain and hindbrain differentiation conditions.

The authors have substantially improved the manuscript by addressing the majority of the three reviewers' comments. While the authors describe an increase in absolute numbers of TH+ cells in their grafts (7,488 TH-positive cells per 100,000 cells grafted vs. hESCs-derived DA neurons 4,250; Fig 7f), others (Kim et al, bioRxiv, 2023) have recently achieved further increases in TH+ cell survival (13,667 (mean) \pm 3,590 (S.E.M) NURR1::GFP dopamine neurons compared to 2,955 (mean) \pm 435 (S.E.M) in wild-type neurons) by targeting TNF-NFkB-p53 signaling. Importantly, in this report, Kim et al show that such an improvement in TH+ cell survival can also be achieved through non-genetic means by pharmacological intervention. Therefore, while the authors show an increase in dopamine neuron differentiation using both midbrain and hindbrain differentiation strategies, I am uncertain of the benefits of using LR-USCs for regenerative medicine purposes.

Reviewer #3 (Remarks to the Author):

The authors were very responsive to the review. This manuscript has been improved. Therefore, I have no more requests to the authors.

REVIEWERS' COMMENTS

Reviewer #1 (Remarks to the Author):

The authors added transplantation experiments with cells differentiated by a midbrain-inducing protocol in the revised manuscript. The results showed a higher yield of DA neurons and faster recovery with the 4X-cells, and only rats with 4X-cells showed improvements in the cylinder test when only 125,000 cells were transplanted.

In the revised Figure 6, the authors showed the advantage of 4X-graft in the brain of 6-OHDA lesioned rats. There are more TH-positive cells in the 4X-cell graft, but the ratio of TH in HNA is not different from the control group. These results suggest that the 4X-cells survived and/or proliferated more than the control cell line. Cell survival in the brain is one of the critical issues to be solved in regenerative therapy, so the results benefit this research field. I recommend adding some interpretations of the mechanism of the better survival of 4X-cells in the brain in the discussion.

Overall, the revised manuscript shows more clearly the advantage of 4X-cells. I am almost satisfied with the answers by the authors, except for one point mentioned above.

Response:

We thank the reviewer for the comments and suggestion.

Our interpretation is that incorrectly specified cells do not survive as well as mesDA neurons do when transplanted into the striatum. Previous papers that have transplanted hindbrain patterned cells into the striatum have also shown small grafts. Therefore, we have now included in the discussion the sentences:

Interestingly, hindbrain patterned H9 cells generated small grafts, which has previously been reported; however, the percentage of DA neurons in the H9 grafts were similar to 4X, which was in contrast to what we saw in vitro. These results indicate that despite the hindbrain specification the few DA neurons in the H9 grafts preferentially survived in vivo, suggests that the denervated striatum is more permissive for the development and survival of TH-positive neurons than in vitro.